# Diffusion-based Cumulative Adversarial Purification for Vision Language Models

## Abstract

Vision Language Models (VLMs) have shown remarkable capabilities in multimodal understanding, yet their susceptibility to adversarial perturbations poses a significant threat to their reliability in real-world applications. Despite often being imperceptible to humans, these perturbations can drastically alter model outputs, leading to erroneous interpretations and decisions. This paper introduces DiffCAP, a novel diffusion-based purification strategy that can effectively neutralize adversarial corruptions in VLMs. We theoretically establish a certified recovery region in the forward diffusion process and meanwhile quantify the convergence rate of semantic variation with respect to VLMs. These findings manifest that adversarial effects monotonically fade as diffusion unfolds. Guided by this principle, DiffCAP leverages noise injection with a similarity threshold of VLM embeddings as an adaptive criterion, before reverse diffusion restores a clean and reliable representation for VLM inference. Through extensive experiments across six datasets with three VLMs under varying attack strengths in three task scenarios, we show that DiffCAP consistently outperforms existing defense techniques by a substantial margin. Notably, DiffCAP significantly reduces both hyperparameter tuning complexity and the required diffusion time, thereby accelerating the denoising process. Equipped with theorems and empirical support, DiffCAP provides a robust and practical solution for securely deploying VLMs in adversarial environments.

Warning: This paper contains images or texts that may be considered offensive.

## 1 Introduction

Vision language models (VLMs) have exhibited impressive performance in a diverse range of multimodal understanding tasks (Radford et al., 2021; Lu et al., 2019; Jia et al., 2021; Alayrac et al., 2022), empowering numerous real-world applications such as image-grounded text generation (e.g., image captioning and visual question answering) (Li et al., 2020; Mokady et al., 2021; Li et al., 2022b) and zero-shot classification (Radford et al., 2021; Zhai et al., 2022; Alayrac et al., 2022; Zhai et al., 2023). However, their inherent susceptibility to adversarial perturbations presents a critical challenge (Zhao et al., 2023; Qi et al., 2024; Zhang et al., 2022). These perturbations are usually designed to be imperceptible to humans, but when added to natural images, can deceive models into making incorrect predictions, severely undermining their reliability and effectiveness (Goodfellow et al., 2014; Madry et al., 2017; Carlini & Wagner, 2017). Adversarial vulnerability is especially concerning as malicious actors may exploit these ML systems to spread misinformation or fraudulent activities (Wu et al., 2024a), highlighting the urgent need for robust defensive strategies (Jin et al., 2024; Liu et al., 2024a).

To mitigate this threat, significant research efforts have focused on adversarial defenses specifically designed for VLMs (Liu et al., 2024a; Weng et al., 2025). A dominant direction in this field is adversarial training, which fine-tunes models using adversarially perturbed data to enhance robustness. For instance, recent approaches such as RobustCLIP (Schlarmann et al., 2024) have leveraged supervised adversarial fine-tuning to fortify VLMs against specific attack types. Although effective within their training scope, these methods exhibit significant limitations, particularly poor generalization to novel, unseen attacks (Dolatabadi et al., 2022; Laidlaw et al., 2020) and substantial computational overhead associated with continuous retraining and fine-tuning procedures (Wong et al., 2020; Andriushchenko & Flammarion, 2020).

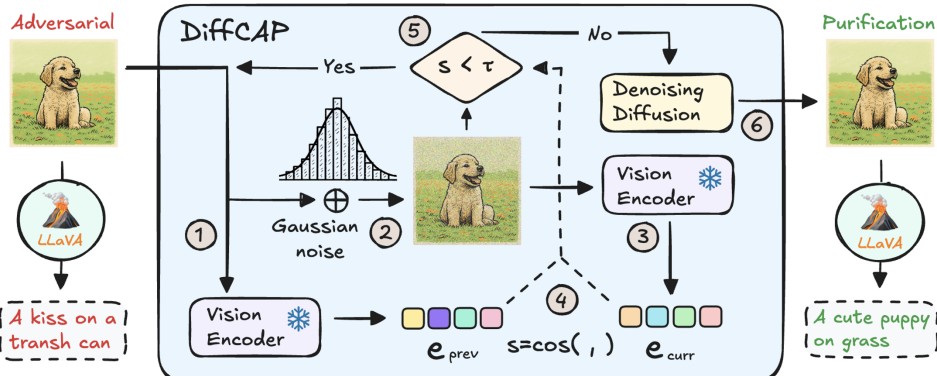

Figure 1: Overview of the DiffCAP Pipeline: An adversarial input is cumulatively processed through steps 1 to 5; if a stopping condition is not satisfied at step 5, the process restarts from step 1, otherwise the purification output is generated at step 6. See Alg. 1 for details.

In contrast, adversarial purification (Yoon et al., 2021; Nie et al., 2022) emerges as a promising alternative that does not require the expensive adversarial fine-tuning of models. Techniques like DiffPure (Nie et al., 2022) have demonstrated the feasibility of purification approaches by directly removing adversarial perturbations from input data using generative models, such as diffusion models (Song et al., 2020a;b; Yang et al., 2023; Song & Ermon, 2019). These methods maintain a high generalization capability to unseen attacks without necessitating modifications to the underlying VLMs, thus preserving original model performance on benign inputs. Despite these advantages, current generative model-based purification techniques suffer from substantial slowdown at inference time, hindering their practical deployment, particularly for real-time scenarios involving large VLMs.

To address these critical shortcomings, we propose DiffCAP, abbreviated for *Diffusion-based Cumulative Adversarial Purification*, the first adversarial purification strategy specifically designed for VLMs. DiffCAP leverages a novel mechanism that dynamically identifies the minimal necessary diffusion time, effectively balancing purification efficacy and computational efficiency. Our method cumulatively injects random Gaussian noise into adversarially perturbed images until the embeddings of two consecutively noised images converge to a predefined similarity threshold, indicating the potential neutralization of adversarial perturbations. A diffusion model subsequently denoises this stabilized image, enabling the recovery of a clean and interpretable image for VLM inference.

**Contribution and novelty.** We summarize as following points:

- We formulate a certified recovery region during the forward diffusion process, with the VLM acting as a zero-shot classifier (Thm. 1). Furthermore, we quantify the convergence rate of VLM-encoded semantic between adjacent forward diffusion steps (Thm. 2). These two theorems reveal that adversarial perturbations can be counteracted after sufficient diffusion steps, with the embedding change diminishing monotonically as diffusion progresses.

- DiffCAP's per-input minimized diffusion time resolves a key limitation of previous diffusion-based purification works, including DiffPure: their reliance on a fixed diffusion time, which is suboptimal for inputs of varying difficulty and requires hyperparameter tuning.

- We conduct comprehensive experiments across three popular VLMs, six diverse datasets, multiple perturbation strengths, and various multimodal tasks, including image captioning, visual question answering, and zero-shot classification. The results demonstrate that DiffCAP consistently outperforms existing state-of-the-art defenses by a considerable margin.

## 2 PRELIMINARIES

This section provides the background and preliminary definitions of vision encoders, including their use for both CLIP and Vision LLMs, as well as continuous-time diffusion models. Detailed related work discussions are supplemented in Appx. B.

## 2.1 VISION ENCODER

**CLIP.** Contrastive Language-Image Pre-training (CLIP) (Radford et al., 2021) consists of a vision encoder $\boldsymbol{\phi} : \mathbb{R}^d \to \mathbb{R}^m$ and a text encoder $\boldsymbol{\psi} : \mathbb{R}^{d'} \to \mathbb{R}^m$. The vision encoder and the text tokenizer have the same embedding dimension. We define a zero-shot classifier $h$. For a $K$-class task, text prompts $\boldsymbol{t}_k$, such as "A photo of <class $k$>", are generated for $k = 1, \ldots, K$. The classifier $h$, determined by $\boldsymbol{\phi}$ and $\boldsymbol{\psi}$, calculates logits for an input image $\boldsymbol{x}$ via the cosine similarity between the image embedding and each prompt embedding:

$$h_k(\boldsymbol{x}) = \cos(\boldsymbol{\phi}(\boldsymbol{x}), \boldsymbol{\psi}(\boldsymbol{t}_k)) = \left\langle \frac{\boldsymbol{\phi}(\boldsymbol{x})}{\|\boldsymbol{\phi}(\boldsymbol{x})\|_2}, \frac{\boldsymbol{\psi}(\boldsymbol{t}_k)}{\|\boldsymbol{\psi}(\boldsymbol{t}_k)\|_2} \right\rangle. \tag{1}$$

**Vision LLMs.** Vision LLMs, such as LLaVA (Liu et al., 2024c; 2023a;b) and MiniGPT (Zhu et al., 2023), consist of a vision encoder $\boldsymbol{\phi} : \mathbb{R}^d \to \mathbb{R}^m$, a text tokenizer, and a language model. The vision encoder and the text tokenizer have the same embedding dimension. When feeding the VLM with an image and the instruction, a vision encoder transforms this image to hidden embeddings of dimension $m$. The text tokenizer first tokenizes the instruction into tokens, and then looks up the embedding matrix to get its $m$-dimensional embedding. The image embedding and the instruction embedding are concatenated and then fed into a language model to generate a description. The vision encoder can be the vision encoder in CLIP (Radford et al., 2021), and the instruction prompt is usually like "Describe this image in detail".

## 2.2 DIFFUSION MODEL

In this section, we briefly introduce the continuous-time diffusion models (Song et al., 2020b). Let $p(\boldsymbol{x})$ represent the underlying, unknown distribution for data points $\boldsymbol{x} \in \mathbb{R}^d$. The core idea of diffusion models is to progressively transform samples from $p(\boldsymbol{x})$ into Gaussian noise. Formally, the transformation can be formulated by the forward diffusion process $\{\boldsymbol{x}(t)\}_{t \in [0,1]}$, which is governed by the following stochastic differential equation (SDE) in the time interval $[0, 1]$:

$$\mathrm{d}\boldsymbol{x} = \boldsymbol{f}(\boldsymbol{x}, t)\mathrm{d}t + g(t)\mathrm{d}\boldsymbol{w}(t), \tag{2}$$

where $\boldsymbol{f} : \mathbb{R}^d \times \mathbb{R} \to \mathbb{R}^d$ stands for the drift function, $g : \mathbb{R} \to \mathbb{R}$ is the diffusion function, and $\boldsymbol{w}(t) \in \mathbb{R}^d$ denotes the Brownian motion. Note that the diffusion process starts with $\boldsymbol{x}(0)$ drawn from the underlying data distribution $p(\boldsymbol{x})$.

The distribution of $\boldsymbol{x}(t)$ at any time $t$ is $p_t(\boldsymbol{x})$, with the initial data distribution being $p_0(\boldsymbol{x}) = p(\boldsymbol{x})$. The functions $\boldsymbol{f}(\boldsymbol{x}, t)$ and $g(t)$ are chosen carefully so that as $t$ approaches 1, the distribution $p_1(\boldsymbol{x})$ closely resembles a standard $d$-dimensional Gaussian distribution, $\mathcal{N}(\boldsymbol{0}, \boldsymbol{I}_d)$. We follow the Variance Preserving (VP) SDE (Song et al., 2020b), where the drift and diffusion coefficients are $\boldsymbol{f}(\boldsymbol{x}, t) = -\frac{1}{2}\beta(t)\boldsymbol{x}$ and $g(t) = \sqrt{\beta(t)}$ respectively. Here, $\beta(t)$ is a function that controls the noise level over time.

To generate new samples, one must reverse the diffusion process. This is achieved by solving a corresponding reverse-time SDE of Eq. (2):

$$\mathrm{d}\hat{\boldsymbol{x}} = \left[\boldsymbol{f}(\hat{\boldsymbol{x}}, t) - g(t)^2 \nabla_{\hat{\boldsymbol{x}}} \log p_t(\hat{\boldsymbol{x}})\right] \mathrm{d}t + g(t)\mathrm{d}\bar{\boldsymbol{w}}. \tag{3}$$

The generation process starts with drawing an initial sample $\hat{\boldsymbol{x}}(1)$ from the standard Gaussian distribution $\mathcal{N}(\boldsymbol{0}, \boldsymbol{I}_d)$. Then, by integrating this SDE from $t = 1$ down to $t = 0$, the noisy sample $\hat{\boldsymbol{x}}(t)$ is progressively denoised. The goal is for the final output $\hat{\boldsymbol{x}}(0)$ to be a sample from the original data distribution $p_0(\boldsymbol{x})$. However, the score function $\nabla_{\boldsymbol{x}} \log p_t(\boldsymbol{x})$ in Eq. (3) is usually intractable. In practice, we use a neural network, denoted by $\boldsymbol{s}_{\boldsymbol{\theta}}(\boldsymbol{x}, t)$ and parameterized by $\boldsymbol{\theta}$, to approximate the score function (Song et al., 2020b; Kingma et al., 2021).

## 3 METHODOLOGY

In this section, we introduce DiffCAP, a purification mechanism that leverages forward diffusion dynamics and semantic stability to remove adversarial perturbations in VLMs. When we pass an

---

**Algorithm 1** DiffCAP: Diffusion-based Cumulative Adversarial Purification

---

**Require:** Adversarially perturbed input image $\boldsymbol{x}_{\mathrm{adv}}$; An image encoder $\boldsymbol{\phi}(\cdot)$ (e.g., from VLM); Similarity threshold $\tau$; Pretrained diffusion denoiser $D(\cdot)$; Maximum number of forward diffusion steps $T$.

**Ensure:** Purified image $\boldsymbol{x}_{\mathrm{clean}}$.

1: Initialize step counter $t \leftarrow 0$.
2: Set initial image $\boldsymbol{x}_0 \leftarrow \boldsymbol{x}_{\mathrm{adv}}$.
3: Calculate initial embedding $\boldsymbol{e}_{\mathrm{prev}} \leftarrow \boldsymbol{\phi}(\boldsymbol{x}_0)$.
4: **while** $t < 1$ **do**                                       ▷ Iteratively inject noise and check stability
5:     Inject noise into the images based on Eq. (5) to obtain $\boldsymbol{x}_{t+1/T}$.
6:     Calculate current embedding $\boldsymbol{e}_{\mathrm{curr}} \leftarrow \boldsymbol{\phi}(\boldsymbol{x}_{t+1/T})$.
7:     **if** $\cos(\boldsymbol{e}_{\mathrm{curr}}, \boldsymbol{e}_{\mathrm{prev}}) \geq \tau$ **then**                 ▷ Check if embeddings have stabilized
8:         **break**                                           ▷ Exit loop, stabilization reached
9:     **end if**
10:    Update previous embedding: $\boldsymbol{e}_{\mathrm{prev}} \leftarrow \boldsymbol{e}_{\mathrm{curr}}$, update $t \leftarrow t + 1/T$.
11: **end while**
12: Set the stabilized (but potentially noisy) image $\boldsymbol{x}_{\mathrm{stable}} \leftarrow \boldsymbol{x}_t$.
13: Denoise the stabilized image using the diffusion model: $\boldsymbol{x}_{\mathrm{clean}} \leftarrow D(\boldsymbol{x}_{\mathrm{stable}})$.
14: **return** $\boldsymbol{x}_{\mathrm{clean}}$.

---

adversarial image $\boldsymbol{x}_{\mathrm{adv}}$ into the diffusion model, $\boldsymbol{x}(t)$ follows a forward diffusion process governed by the variance-preserving stochastic differential equation:

$$\mathrm{d}\boldsymbol{x}(t) = -\tfrac{1}{2}\beta(t)\boldsymbol{x}(t)\mathrm{d}t + \sqrt{\beta(t)}\,\mathrm{d}\boldsymbol{w}(t), \quad \boldsymbol{x}(0) = \boldsymbol{x}_{\mathrm{adv}}, \tag{4}$$

$\beta(t) > 0$ is a smooth noise schedule, and $\boldsymbol{w}(t)$ is a standard Wiener process. This process has a closed-form solution at time $t$ given by:

$$\boldsymbol{x}(t) = \sqrt{\alpha(t)}\,\boldsymbol{x}_{\mathrm{adv}} + \sqrt{1 - \alpha(t)}\,\boldsymbol{\epsilon}, \quad \boldsymbol{\epsilon} \sim \mathcal{N}(\boldsymbol{0}, \boldsymbol{I}_d), \tag{5}$$

where $\alpha(t) = \exp\left(-\int_0^t \beta(s)\,\mathrm{d}s\right)$. Our method builds upon the insight from randomized smoothing (Cohen et al., 2019): adding Gaussian noise to adversarial inputs can recover the original predictions with high probability. We need the following basic assumptions to facilitate our analysis.

**Assumption 1** (Scale Invariance). *We assume that the classifier $h$ (defined in Eq. (1)) is scale-invariant: for any scalar $\lambda > 0$ and any input $\boldsymbol{x} \in \mathbb{R}^d$, we have $h(\lambda \boldsymbol{x}) = h(\boldsymbol{x})$.*

**Remark 1.** *Asm. 1 is standard in deep learning theory and practically justified for models as we can always scale the input (Zhang et al., 2020; Wu et al., 2024b).*

We now present our main result that establishes a certified recovery region under forward diffusion for the adversarial image. The proof can be found at Appx. C.1.

**Theorem 1** (Certified recovery region under forward diffusion). *Let $h : \mathbb{R}^d \to [K]$ be the classifier[1]. Let $\boldsymbol{x}_{adv} = \boldsymbol{x} + \boldsymbol{\epsilon}_{adv}$ be an adversarial example with perturbation $\boldsymbol{\epsilon}_{adv}$, and let $\boldsymbol{x}(t)$ be the solution to the forward diffusion process defined in Eq. (5) with a linear noise schedule $\beta(t) = \beta_{\min} + (\beta_{\max} - \beta_{\min})t$. Suppose there exists $\underline{p_1}, \overline{p_2}, k_1$ such that for all $t \in [0, 1]$*

$$\mathbb{P}(h(\boldsymbol{x} + \boldsymbol{\epsilon}'(t)) = k_1) \geq \underline{p_1} > \overline{p_2} \geq \max_{k \neq k_1} \mathbb{P}(h(\boldsymbol{x} + \boldsymbol{\epsilon}'(t)) = k), \tag{6}$$

*where $\boldsymbol{\epsilon}'(t) \sim \mathcal{N}(\boldsymbol{0}, \frac{1-\alpha(t)}{\alpha(t)}\boldsymbol{I}_d)$. Define*

$$t_{\min} = \frac{2M}{\sqrt{\beta_{\min}^2 + 2(\beta_{\max} - \beta_{\min})M} + \beta_{\min}}, \quad \text{where } M := \log\left(1 + \left(\frac{2\|\boldsymbol{\epsilon}_{adv}\|_2}{\boldsymbol{\phi}^{-1}(\underline{p_1}) - \boldsymbol{\phi}^{-1}(\overline{p_2})}\right)^2\right).$$

*Then, under Asm. 1, when $\beta_{\min} \geq M$, for all $t_{\min} \leq t \leq 1$, we have $\arg\max_{k \in [K]} \mathbb{P}(h(\boldsymbol{x}(t)) = k) = k_1$, i.e., the adversarial example is classified as its original label $k_1$ after sufficient forward diffusion.*

---

[1]Our analysis can be extended to general VLM tasks and we provide justification in Appx. C.4.

---

**Algorithm 2** Adaptive Similarity Threshold ($\tau$) Calculation

---

**Require:** The dataset of clean-adversarial image pairs $D_{\text{pairs}} = \{(\boldsymbol{x}_{\text{clean}}, \boldsymbol{x}_{\text{adv}})\}$; The embedding function $\phi(\cdot)$. Maximum number of forward diffusion steps $T$.
**Ensure:** The similarity threshold $\tau$.
 1: Initialize step counter $t \leftarrow 0$.
 2: **for all** pair $(\boldsymbol{x}_{\text{clean}}, \boldsymbol{x}_{\text{adv}}) \in D_{\text{pairs}}$ **do**
 3:      Set initial image $\boldsymbol{x}_{0,\text{clean}} \leftarrow \boldsymbol{x}_{\text{clean}}, \boldsymbol{x}_{0,\text{adv}} \leftarrow \boldsymbol{x}_{\text{adv}}$.
 4:      Calculate initial embedding $\boldsymbol{e}_{\text{prev,clean}} \leftarrow \phi(\boldsymbol{x}_{0,\text{clean}}), \boldsymbol{e}_{\text{prev,adv}} \leftarrow \phi(\boldsymbol{x}_{0,\text{adv}})$.
 5: **end for**
 6: **while** $t < 1$ **do**                                  ▷ Iteratively inject noise and check stability
 7:      Initialize total similarity set $S_{\text{clean}} \leftarrow [], S_{\text{adv}} \leftarrow []$.
 8:      **for all** pair $(\boldsymbol{x}_{\text{clean}}, \boldsymbol{x}_{\text{adv}}) \in D_{\text{pairs}}$ **do**
 9:          Inject noise into the images based on Eq. (5) to obtain $\boldsymbol{x}_{t+1/T,\text{clean}}, \boldsymbol{x}_{t+1/T,\text{adv}}$.
10:          Calculate current embedding $\boldsymbol{e}_{\text{curr,clean}} \leftarrow \phi(\boldsymbol{x}_{t+1/T,\text{clean}}), \boldsymbol{e}_{\text{curr,adv}} \leftarrow \phi(\boldsymbol{x}_{t+1/T,\text{adv}})$.
11:          Calculate similarity $s_{\text{clean}} \leftarrow \cos(\boldsymbol{e}_{\text{curr,clean}}, \boldsymbol{e}_{\text{prev,clean}}), s_{\text{adv}} \leftarrow \cos(\boldsymbol{e}_{\text{curr,adv}}, \boldsymbol{e}_{\text{prev,adv}})$.
12:          Add the similarity score to the set $S_{\text{clean}} \leftarrow S_{\text{clean}} \cup \{s_{\text{clean}}\}, S_{\text{adv}} \leftarrow S_{\text{adv}} \cup \{s_{\text{adv}}\}$.
13:      **end for**
14:      **if** $S_{\text{adv}}$ and $S_{\text{clean}}$ are from the same underlying distribution **then**
15:          **break**
16:      **end if**
17:      **for all** pair $(\boldsymbol{x}_{\text{clean}}, \boldsymbol{x}_{\text{adv}}) \in D_{\text{pairs}}$ **do**
18:          Update previous embedding: $\boldsymbol{e}_{\text{prev,clean}} \leftarrow \boldsymbol{e}_{\text{curr,clean}}, \boldsymbol{e}_{\text{prev,adv}} \leftarrow \boldsymbol{e}_{\text{curr,adv}}$.
19:      **end for**
20:      Update $t \leftarrow t + 1/T$.
21: **end while**
22: **return** $\tau \leftarrow \text{mean}(S_{\text{clean}})$.

---

**Remark 2.** *Thm. 1 indicates that an adversarially perturbed image with added noise will eventually be classified correctly. Then, we can expect to use the diffusion model to remove the added noise to obtain the clean image.*

Upon this point, we start to analyze the dynamics in the VLM embedding space during the forward diffusion process. In Thm. 1, we prove a certified recovery region, indicating the local smoothness of $\phi(\boldsymbol{x}(t))$ for $t \geq t_{\min}$. Therefore, we pose a local Lipschitz assumption after time $t_{\min}$:

**Assumption 2.** *We assume $\phi$ is $L-$Lipschitz for $t \geq t_{\min}$*

$$\|\phi(\boldsymbol{x}(t)) - \phi(\boldsymbol{x}(t'))\|_2 \leq L\|\boldsymbol{x}(t) - \boldsymbol{x}(t')\|_2 \quad \forall t, t' \geq t_{\min}.$$

**Lemma 1.** *Under the same setting as in Thm. 1 and Asm. 2, let $\boldsymbol{x}(t)$ be defined as in Eq. (5). Then for any $t_1, t_2 \in [0, 1]$, as $t_1, t_2 \to 1$, we have: $\mathbb{E}\left[\|\phi(\boldsymbol{x}(t_1)) - \phi(\boldsymbol{x}(t_2))\|_2\right] \to 0$.*

The proof is deferred to Appx. C.2. Lemma 1 shows that embeddings converge during forward diffusion, which motivates our stopping criterion based on similarity of VLM latent representations. We quantify this convergence rate in the following theorem:

**Theorem 2.** *Let $\boldsymbol{x}(t)$ be defined as in Eq. (5). Then for small $\delta > 0$,*

$$\mathbb{E}\left[\|\phi(\boldsymbol{x}(t)) - \phi(\boldsymbol{x}(t + \delta))\|_2\right] = O\left(L \cdot \delta \cdot \beta(t)\sqrt{\frac{\alpha(t)}{1 - \alpha(t)}}\right), \text{ for } t \in [t_{\min}, 1 - \delta], \quad (7)$$

*where $\alpha(t) = \exp\left(-\int_0^t \beta(s)\,\mathrm{d}s\right)$. Moreover, by using a common linear noise schedule $\beta(t) = \beta_{\min} + (\beta_{\max} - \beta_{\min})t$ with $\beta_{\min} > 0$ and $\beta_{\max} > \beta_{\min}$. Then $\beta(t) \cdot \sqrt{\frac{\alpha(t)}{1-\alpha(t)}}$ is strictly decreasing for all $t \in [t_{\min}, 1)$.*

The proof is deferred to Appx. C.3. Thm. 2 quantifies the semantic change between adjacent forward diffusion steps. The bound in Eq. (7) decreases as $t \to 1$ under the common linear noise schedule.

Consequently, the expected semantic change between adjacent forward diffusion steps diminishes as the process approaches terminal time.

Our derivations inspire a simple yet powerful strategy—inject Gaussian noise until the semantic embedding stabilizes. We call this algorithm DiffCAP, summarized in Alg. 1. The cumulative diffusion continues until the cosine similarity between consecutive VLM embeddings exceeds a threshold $\tau$. A pretrained diffusion model is then applied in reverse to recover a clean image from the stabilized noisy input.

We describe how to select the threshold $\tau$ in Alg. 2. The core idea is to iteratively inject noise into both clean and adversarial images and track the cosine similarity between the embeddings across consecutive steps of noise injection. This is done for a collection of image pairs to reduce randomness. The process continues until the set of similarity scores for clean images and the set for their adversarial counterparts are statistically indistinguishable (i.e., likely from the same underlying distribution). At this point, the algorithm determines that the noise has reached a level where the embeddings' stability is comparable for both types of images. The mean of such a score set delivers the final threshold $\tau$.

## 4 EXPERIMENTS

### 4.1 SETTINGS

**Models, datasets & metrics.** We evaluate DiffCAP across three vision-language tasks: image captioning (IC), visual question answering (VQA), and zero-shot classification (ZSC). For IC and VQA, we adopt two large VLMs—OpenFlamingo (OF) (Awadalla et al., 2023) with 9B parameters and LLaVA-1.5 (Liu et al., 2024b) with 7B parameters. For ZSC, we utilize CLIP (Radford et al., 2021) with 88M parameters as the backbone model. Our experiments are conducted on standard benchmarks: COCO (Lin et al., 2014) and Flickr30k (Plummer et al., 2015) for IC, VQAv2 (Goyal et al., 2017) and TextVQA (Singh et al., 2019) for VQA, and CalTech101 (Li et al., 2022a) and ImageNet1K (Deng et al., 2009) for ZSC. For both adversarial and clean evaluation, we randomly sample 500 images for IC and VQA, while $1,000$ images are chosen for ZSC. We report Consensus-based Image Description Evaluation (CIDEr) (Vedantam et al., 2015) score for IC, VQA accuracy (Antol et al., 2015) for VQA, and top-1 accuracy for ZSC.

**Attacks.** The attack is conducted in the *gray-box* setting, where the adversary can access the gradients of the model but has no knowledge of the defense pipeline. For IC and VQA, we adopt a two-stage attack pipeline following (Schlarmann & Hein, 2023). In the first stage, we apply 100-step Auto-PGD (APGD) attacks (Croce & Hein, 2020) in half-precision using multiple ground-truth captions or answers as supervision. Samples that fall below a predefined performance threshold are excluded from further attacks. In the second stage, we conduct stronger single-precision APGD attacks on the remaining samples. This progressive strategy maximizes adversarial impact while remaining computationally efficient. For ZSC, we follow the AutoAttack framework, employing APGD with cross-entropy loss and targeted Difference of Logits Ratio (DLR) loss with 100 iterations, respectively.

**Baselines.** We compare DiffCAP with two categories of adversarial defense methods. The first category includes adversarially fine-tuned vision encoders. Since both OF and LLaVA adopt CLIP as their vision backbone, we replace their CLIP vision encoder with two robust variants: TeCoA (Mao et al., 2022) and FARE (Schlarmann et al., 2024). TeCoA applies supervised adversarial training, while FARE employs an unsupervised loss. The second category includes purification methods. We consider: JPEG-DL (Salamah et al., 2024), the trainable JPEG compression layer to remove adversarial perturbations; DiffPure (Nie et al., 2022), the first approach leveraging the diffusion process to recover the clean image in the pixel space; CLIPure (Zhang et al., 2025), the latest method that operates directly in the CLIP latent space.

**Hyperparameters.** The algorithms are implemented through PyTorch, and all experiments are conducted on an NVIDIA A100 40G GPU. By default, we use the ViT-B/32 CLIP vision encoder to ensure computational efficiency. We take advantage of the pre-trained diffusion model from (Dhariwal & Nichol, 2021). Following Alg. 2, we determine the threshold $\tau = 0.96$ on subsets comprising 100 random clean-adversarial image pairs from the datasets mentioned above. For all the experiments, we fix the threshold. More experimental details can be found in Appx. D.

Table 1: CIDEr score of two VLMs in IC task on two datasets with clean images and adversarial perturbations of two sizes under different defenses. 2 and 4 with TeCoA and FARE suggest the version that is fine-tuned by $\ell_\infty^{2/255}$ and $\ell_\infty^{4/255}$ bounded adversarial examples, respectively. The best result is in **bold** and the runner-up is underlined.

| Defense | OF-9B | | | | | | LLaVA 1.5-7B | | | | | |
| | COCO | | | Flickr30k | | | COCO | | | Flickr30k | | |
| | clean | $\ell_\infty^{2/255}$ | $\ell_\infty^{4/255}$ | clean | $\ell_\infty^{2/255}$ | $\ell_\infty^{4/255}$ | clean | $\ell_\infty^{2/255}$ | $\ell_\infty^{4/255}$ | clean | $\ell_\infty^{2/255}$ | $\ell_\infty^{4/255}$ |
|---|---|---|---|---|---|---|---|---|---|---|---|---|
| No defense | 79.7 | 1.5 | 1.1 | 60.1 | 0.7 | 0.4 | 115.5 | 4.0 | 3.1 | 77.5 | 1.6 | 1.0 |
| TeCoA[2] (Mao et al., 2022) | 73.5 | 31.6 | 21.2 | 49.5 | 14.1 | 9.5 | 98.4 | 44.2 | 30.3 | 57.1 | 23.2 | 15.3 |
| FARE[2] (Schlarmann et al., 2024) | 79.1 | 34.2 | 19.5 | 57.7 | 16.4 | 8.9 | 109.9 | 53.6 | 31.0 | 71.1 | 29.5 | 17.5 |
| TeCoA[4] (Mao et al., 2022) | 66.9 | 28.5 | 21.6 | 40.9 | 12.0 | 10.3 | 88.3 | 50.9 | 35.3 | 48.6 | 27.9 | 19.5 |
| FARE[4] (Schlarmann et al., 2024) | 74.1 | 30.9 | 22.8 | 51.4 | 15.7 | 10.5 | 102.4 | 57.1 | 40.9 | 61.6 | 31.4 | 22.8 |
| JPEG-DL (Salamah et al., 2024) | 78.2 | 66.1 | 43.9 | 58.8 | 47.6 | 30.7 | 113.3 | 106.4 | 77.2 | 74.8 | 69.6 | 47.9 |
| DiffPure (Nie et al., 2022) | 74.9 | 73.4 | 72.3 | 49.8 | 49.2 | 50.3 | 106.5 | 108.4 | 105.0 | 65.5 | 66.4 | 63.2 |
| CLIPure (Zhang et al., 2025) | 80.8 | 6.6 | 5.3 | 59.3 | 4.7 | 3.5 | 115.1 | 4.9 | 3.4 | 76.9 | 2.1 | 1.5 |
| DiffCAP | **81.4** | **79.3** | **78.4** | 55.6 | **56.7** | **57.2** | **120.4** | **119.6** | **116.9** | 75.0 | **72.7** | **72.1** |

Table 2: VQA accuracy (%) of two VLMs in VQA task on two datasets with clean images and adversarial perturbations of two sizes under different defenses. 2 and 4 with TeCoA and FARE suggest the version that is fine-tuned by $\ell_\infty^{2/255}$ and $\ell_\infty^{4/255}$ bounded adversarial examples, respectively. The best result is in **bold** and the runner-up is underlined.

| Defense | OF-9B | | | | | | LLaVA 1.5-7B | | | | | |
| | TextVQA | | | VQAv2 | | | TextVQA | | | VQAv2 | | |
| | clean | $\ell_\infty^{2/255}$ | $\ell_\infty^{4/255}$ | clean | $\ell_\infty^{2/255}$ | $\ell_\infty^{4/255}$ | clean | $\ell_\infty^{2/255}$ | $\ell_\infty^{4/255}$ | clean | $\ell_\infty^{2/255}$ | $\ell_\infty^{4/255}$ |
|---|---|---|---|---|---|---|---|---|---|---|---|---|
| No defense | 23.8 | 0.0 | 0.0 | 48.5 | 1.8 | 0.0 | 37.1 | 0.5 | 0.0 | 74.5 | 2.9 | 0.0 |
| TeCoA[2] (Mao et al., 2022) | 16.6 | 3.5 | 2.1 | 46.2 | 23.5 | 20.5 | 24.1 | 12.1 | 8.8 | 66.9 | 33.8 | 21.8 |
| FARE[2] (Schlarmann et al., 2024) | 21.6 | 4.1 | 1.9 | 47.0 | 24.0 | 17.2 | 31.9 | 14.7 | 9.1 | 71.7 | 34.9 | 23.0 |
| TeCoA[4] (Mao et al., 2022) | 15.4 | 2.1 | 1.8 | 44.8 | 23.6 | 21.3 | 20.7 | 12.6 | 9.3 | 63.2 | 41.0 | 31.7 |
| FARE[4] (Schlarmann et al., 2024) | 18.6 | 3.4 | 2.9 | 46.1 | 23.6 | 21.0 | 27.6 | 15.8 | 10.9 | 68.3 | 40.7 | 30.5 |
| JPEG-DL (Salamah et al., 2024) | **23.4** | 15.9 | 13.1 | 46.8 | 39.5 | 32.4 | 34.6 | 27.2 | 21.1 | 68.8 | 60.8 | 45.8 |
| DiffPure (Nie et al., 2022) | 13.6 | 13.2 | 13.5 | 45.1 | 43.5 | 43.6 | 20.9 | 22.0 | 22.2 | 67.3 | 65.8 | 66.0 |
| CLIPure (Zhang et al., 2025) | 20.5 | 6.8 | 8.8 | **47.3** | 18.8 | 17.5 | **36.1** | 2.1 | 1.4 | **73.3** | 4.6 | 2.1 |
| DiffCAP | 18.6 | **16.2** | **16.7** | 46.3 | **45.4** | **45.3** | 28.3 | **29.0** | **28.9** | 70.3 | **69.1** | **68.5** |

## 4.2 RESULT ANALYSIS

**Image captioning.** As shown in Tab. 1, VLMs are highly vulnerable to adversarial perturbations: even 2/255 attacks can reduce CIDEr scores close to zero. Adversarial training methods (TeCoA and FARE) provide moderate robustness improvements. However, their effectiveness drops significantly under 4/255 attacks. Notably, TeCoA and FARE also degrade the clean performance, especially on Flickr30k. JPEG-DL shows better robustness than TeCoA and FARE, but remains sensitive to perturbation strength. DiffPure substantially improves robustness, lifting performance under both 2/255 and 4/255 attacks to levels comparable with clean conditions. DiffCAP consistently outperforms all baselines across both VLMs and datasets. For instance, DiffCAP improves CIDEr scores by over 10% with OF on Flickr30k and with LLaVA on COCO compared to DiffPure. Furthermore, DiffCAP maintains or even improves clean performance, demonstrating strong fidelity preservation. Lastly, CLIPure performs poorly in this task. Its limited effectiveness likely stems from token-level misalignment: purifying only the [CLS] token embedding fails to influence generation-related latent tokens, which dominate the captioning process.

**Visual question answering.** The results in Tab. 2 mirror the trends observed in Tab. 1. DiffCAP consistently delivers the strongest performance across all attack settings compared to all baselines in both datasets and VLMs. The improvement is particularly notable with LLaVA on TextVQA, where DiffCAP surpasses DiffPure by over 30%. Remarkably, on the TextVQA dataset, DiffPure performs worse than JPEG-DL, indicating that visual reasoning tasks depend more heavily on fine-grained visual features, which are susceptible to over-smoothing or distortion during purification. This underscores the importance of preserving semantic fidelity when applying generative models for purification. DiffCAP addresses this by dynamically calculating the minimal diffusion time required

Table 3: Evaluation for OF-9B and LLaVA 1.5-7B in IC and VQA tasks on four datasets under clean and adversarial ($\ell_\infty^{8/255}$) conditions, with and without (w/o) DiffCAP defense against adaptive attacks.

| | Dataset | Clean | | APGD | | BPDA | | BPDA + EOT | |
|---|---|---|---|---|---|---|---|---|---|
| | | w/o | with | w/o | with | w/o | with | w/o | with |
| OF | COCO | 90.1 | 92.4 | 4.7 | 91.1 | 27.1 | 79.9 | 29.2 | 83.4 |
| | Flicker30k | 63.9 | 62.7 | 4.9 | 60.5 | 19.1 | 50.6 | 15.5 | 56.5 |
| | TextVQA | 23.1 | 18.6 | 0.6 | 17.6 | 7.1 | 18.2 | 2.3 | 16.0 |
| | VQAv2 | 46.2 | 47.1 | 8.3 | 44.6 | 24.0 | 44.5 | 18.0 | 39.5 |
| LLaVA 1.5 | COCO | 125.9 | 122.2 | 11.3 | 123.4 | 21.9 | 115.9 | 19.5 | 114.9 |
| | Flicker30k | 81.7 | 78.0 | 8.5 | 76.2 | 20.9 | 73.4 | 18.0 | 74.6 |
| | TextVQA | 36.9 | 25.1 | 7.4 | 22.7 | 8.8 | 24.3 | 8.7 | 21.7 |
| | VQAv2 | 74.3 | 69.9 | 23.4 | 67.5 | 25.7 | 65.4 | 27.1 | 66.9 |

Table 4: Top-1 accuracy (%) in ZSC task. We use different CLIP vision encoders for DiffCAP. Numbers in () denote parameters in M.

| | Encoder | clean | $\ell_\infty^{2/255}$ | $\ell_\infty^{4/255}$ |
|---|---|---|---|---|
| CalTech101 | RN50 (102) | 82.8 | 82.2 | 82.5 |
| | RN101 (123) | 83.2 | 83.1 | 81.2 |
| | ViT-B/32 (88) | 82.6 | 81.7 | 80.9 |
| | ViT-B/16 (149) | 83.0 | 82.3 | 80.9 |
| | ViT-L/14 (304) | 82.1 | 82.4 | 81.5 |
| ImageNet1K | RN50 (102) | 84.2 | 82.9 | 80.8 |
| | RN101 (123) | 86.7 | 85.5 | 84.1 |
| | ViT-B/32 (88) | 87.2 | 84.4 | 81.1 |
| | ViT-B/16 (149) | 84.7 | 85.0 | 82.2 |
| | ViT-L/14 (304) | 85.5 | 83.2 | 81.4 |

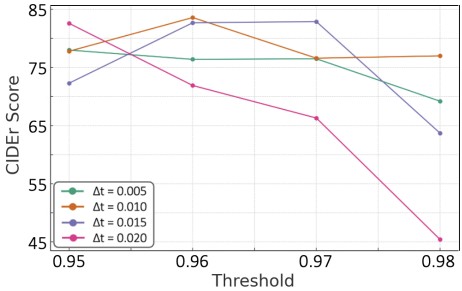 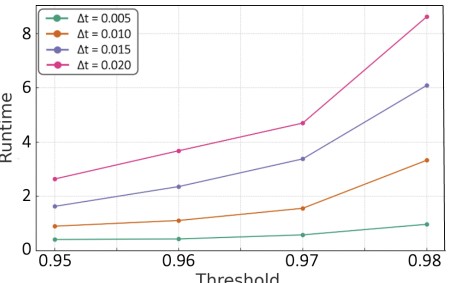

Figure 2: CIDEr score and running time (in seconds) per image with varying thresholds $\tau$ and diffusion step sizes ($\Delta t$) for DiffCAP. The evaluation is based on the IC task under $\ell_\infty^{2/255}$ attack.

to remove adversarial noise for each individual image, thereby achieving a better trade-off between robustness and feature integrity for multi-hop reasoning tasks.

**Adaptive attack.** We also evaluate DiffCAP in a *white-box* setting, where the adversary has full knowledge of the deployed defense mechanism. Tab. 3 presents the detailed evaluations for image captioning (IC) and visual-question answering (VQA) tasks across various attack configurations, comparing performance with and without DiffCAP defense. Even under an increased attack budget (8/255), DiffCAP maintains high fidelity on clean inputs, with only an average performance drop of 3.3 points. Under APGD (Croce & Hein, 2020) attacks, when the adversary is unaware of DiffCAP's existence, it successfully restores the performance of VLMs on different datasets to levels closely matching their clean baselines, showing only a modest average degradation of 4.8 points.

In scenarios where the adversary bypasses gradient obfuscation through backward pass differentiable approximation (BPDA) (Athalye et al., 2018a) and simulates stochasticity via expectation over transformations (EOT) (Athalye et al., 2018b), DiffCAP continues to demonstrate strong resilience. The best-case performance degradation relative to clean conditions is only 1.7 points (OF-VQAv2) by BPDA without EOT and 6.7 points (OF-COCO) with EOT. The corresponding worst-case performance reductions are observed as 13.3 points (OF-Flickr30k) and 15.2 points (LLaVA-TextVQA), respectively. These results elucidate the inherent uncertainty of DiffCAP's image-adaptive diffusion step calculation, which determines the minimal purification for individual adversarial examples based on semantic convergence during the diffusion process. Such a dynamic strategy significantly prevents trivial gradient approximations and random regressions from circumventing its defense, enhancing adversarial robustness against adaptive attacks of prohibitively high time complexity.

**Ablation study.** We conduct systematic ablation studies to validate the utility of DiffCAP. Tab.4 presents results obtained by replacing the vision encoder in DiffCAP with different CLIP backbones. The results illustrate that DiffCAP is largely insensitive to the choice of vision encoder, maintaining robustness across all variants. To validate the effectiveness of the adaptive similarity threshold calculation described in Alg. 2, we conduct an ablation study over different threshold values $\tau$ and

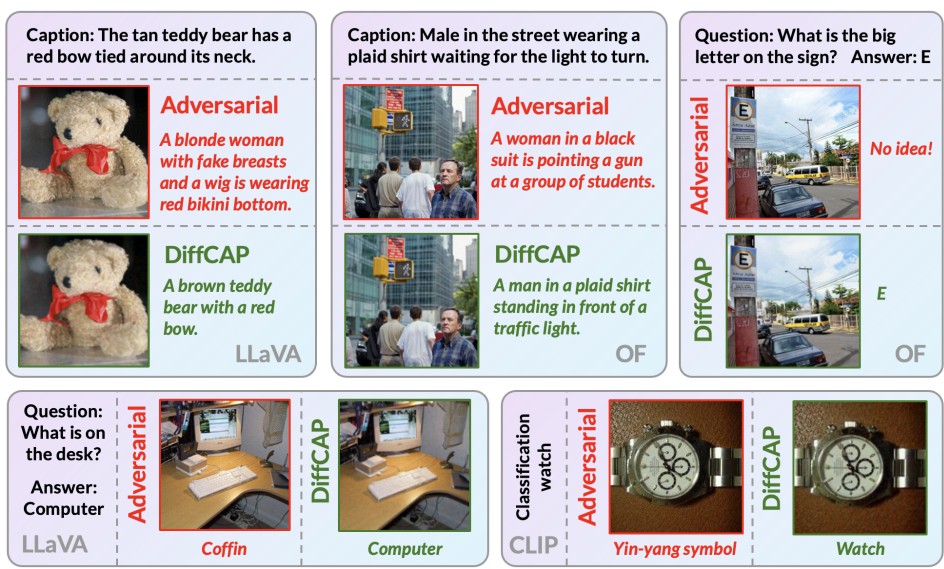

Figure 3: Adversarial examples and their DiffCAP purified outcomes under different tasks. Ground-truth labels are shown in black text. VLMs used for inference are shown in gray text.

diffusion step sizes $\Delta t$. Fig. 2 displays the results by OF on the COCO dataset. We observe that setting the threshold to $0.96$ achieves the best overall robustness in terms of CIDEr score across a range of step sizes. A higher threshold generally leads to more diffusion steps, increasing time for reverse diffusion sampling. In practice, we find that a step size of $0.01$ offers the best trade-off between performance and computational cost.

**Further Discussion.** In Appx. D, we supplement additional experimental results on the ZSC task, $16/255$ attacks, a more complex model MiniGPT 4 (Zhu et al., 2023) under a more sophisticated attack AttackVLM (Zhao et al., 2023), robustness to visual hallucination (Li et al., 2023) and jailbreaking (Qi et al., 2024), perceptual quality, $\ell_2$ attacks, computational overhead, and under-/over-purification. In Appx. E, we further attach a systematic analysis of DiffCAP's critical hyperparameters, including the threshold (across calibration subsets, domain shifts, task transformations, and CLIP variants), the step size (on robustness-fidelity trade-off), and the noise scheduler.

**Efficiency.** For the $\ell_\infty^{2/255}$ attack on the COCO dataset with OF-9B, DiffCAP demonstrates substantial efficiency gain over DiffPure in IC task. After a one-time calibration of threshold $\tau$ ($\sim 3.4$ seconds) by Alg. 2, the average purification time is only $\sim 1.1$ seconds per image for DiffCAP, in contrast to $\sim 2.3$ seconds for DiffPure. The embedding extraction and the Gaussian noise injection consume only $\sim 6$ and $\sim 4$ milliseconds per iteration, respectively. Since the reverse denoising dominates the runtime, the additional overhead from embedding comparisons hardly offsets the runtime savings achieved through the reduced $\sim 2/3$ diffusion steps over DiffPure (illustrated by Fig. 4), ensuring DiffCAP's practicality for real-time deployment scenarios.

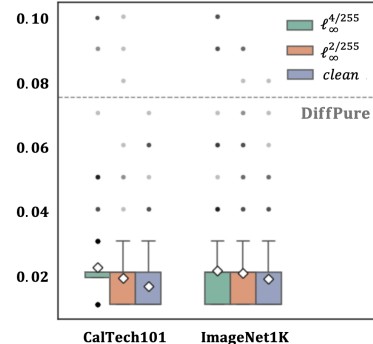

Figure 4: The box plot of DiffCAP's diffusion time $t$ ($y$-axis) before existing noise injection loop. The dashline ($t = 0.075$) is the noise injection time tuned on $\ell_\infty$ attack reported by DiffPure paper. DiffCAP requires significantly smaller diffusion time than DiffPure. The dots mark outliers and rhombuses mark mean values.

Fig. 3 showcases the purification consequence of Diff-CAP in IC, VQA, and ZSC scenarios. As a generative adversarial purification method, it introduces no noticeable degradation in fidelity. DiffCAP prominently mitigates the tension between robustness, efficiency, and image quality, establishing a new state-of-the-art among both purification- and training-based defenses for VLMs.

## 5 CONCLUSION

In conclusion, this paper proposes DiffCAP, an efficient and theoretically inspired defense strategy for VLMs, supported by a certified recovery region and descending semantic change in forward diffusion. By leveraging cumulative Gaussian noise injection and a VLM embedding similarity-based stopping criterion, DiffCAP dynamically identifies the minimal purification steps required before denoising, substantially reducing computational overhead while maintaining high fidelity. DiffCAP consistently outperforms existing defenses across diverse tasks, VLMs, and datasets empirically.

### ETHICS STATEMENT

This work complies with the ICLR Code of Ethics. It does not involve human/animal subjects, and raises no concerns regarding bias/privacy/legal/integrity issues. As a defense mechanism against harmful adversarial attacks, our method could potentially be exploited to develop more advanced adaptive attacks in the future. Nevertheless, such a possibility should not hinder progress on building more robust and trustworthy AI systems. We believe the contributions of this work have a positive impact on the development of our community.

### REPRODUCIBILITY STATEMENT

We provide the clear assumptions and a complete proof of the proposed theorems and lemmas in Sec. 3 and Appx. C, respectively. All used datasets/VLMs, pretrained diffusion models, and applied adversarial attacks in this work are publicly available. Our code will also be released upon the acceptance. In the meantime, the relevant experimental setups are fully described in paper Sec. 4.1 and Appx. D.1, together with the step-by-step algorithm flow Alg. 1 and Alg. 2, ensuring that the results can be manually reproduced.

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

CONTENTS OF THE APPENDIX

We organize the appendix as follows:

- In Appx. A, we declare the LLM usage, limitations, border impact of this paper.
- In Appx. B, we elaborate detailed related adversarial attack and defense works.
- In Appx. C, we provide complete proofs for Thm. 1 and 2 and Lemma 1.
- In Appx. D, we supply more implementation details and experiment results. We further explore the potential of applying DiffCAP to boarder safety alignment challenges.
- In Appx. E, we attach a comprehensive analysis of the key hyperparameters of DiffCAP.

## A  AUTHOR STATEMENTS

**The LLM usage.** We only use the ChatGPT-4o (OpenAI, 2024) to rectify writing errors paragraph by paragraph. Our prompt is "*Please only revise the necessary part of the following paragraph if there are any incorrect grammar, unclear syntax, or unacademic expression:* [our paragraph]."

**Future work.** While our method is highly effective in the vision modality, an important limitation is its current reliance on image-based diffusion models. Extending the cumulative purification framework to text or multimodal diffusion processes remains an open direction, potentially broadening its applicability to adversarial text modifications or joint vision-text threats.

**Social impact.** With the swift application of VLMs, the risk of adversarial attacks has become a critical concern. This paper proposes DiffCAP, an adversarial purification method that can improve robustness without retraining the model, which may enlarge its usability in application scenarios, such as autonomous driving based on VLMs. While maintaining a remarkable defense effect, this method greatly reduces the diffusion steps and hyperparameter adjustments, which promotes the safe and fast implementation for defending pre-trained large models. However, any single defense mechanism may fail in the presence of new attacks, so maintaining a diverse and regularly tested defense strategy is essential.

## B  RELATED WORK

**Perturbation-based attacks.** Perturbation-based attacks cause models to make incorrect predictions by introducing small, often imperceptible, alterations to the input data (Chakraborty et al., 2021; Huang et al., 2017; Chakraborty et al., 2018). These attacks commonly leverage gradient-based methods to find the most vulnerable parts of an input, then craft perturbations to maximize the model's loss. A classic illustration is adversarial image attacks, where minute pixel modifications, invisible to humans, can trick a model into misclassifying an image (Goodfellow et al., 2014; Madry et al., 2017).

**Adversarial training.** Adversarial training employs optimization techniques to bolster model robustness and safety alignment. TeCoA (Mao et al., 2022) applies supervised adversarial fine-tuning on ImageNet, while FARE (Schlarmann et al., 2024) leverages an unsupervised adversarial fine-tuning approach using an embedding loss on PGD-perturbed (Madry et al., 2017) inputs to enhance CLIP vision encoder robustness and zero-shot performance. Addressing challenges in such methods, Hossain et al. (Hossain & Imteaj, 2024b) developed Sim-CLIP, which integrates a Siamese architecture with a cosine similarity loss to align clean and perturbed representations, incorporating a stop-gradient mechanism for efficient training without negative samples. This was further extended by Sim-CLIP+ (Hossain & Imteaj, 2024a), which tailors the cosine similarity loss and stop-gradient mechanism to defend VLMs against advanced optimization-based jailbreak attacks, preventing symmetric loss collapse while maintaining computational efficiency.

**Adversarial purification.** Adversarial purification techniques (Shi et al., 2021; Yoon et al., 2021) offer a distinct defense paradigm by employing generative models to sanitize images from adversarial perturbations (Samangouei et al., 2018; Hill et al., 2021). The main advantage is its "plug-in" simplicity to address new threats without the need to retrain vision models—since the adversarial images

being sanitized independently of attack specifics and the vision models. However, this adaptability used to be constrained by weaker performance compared to adversarial training methods (Croce & Hein, 2020). The vulnerability becomes especially clear when faced with adaptive attackers who have knowledge of the defense system (Athalye et al., 2018a; Tramer et al., 2020), a problem generally rooted in the inherent weaknesses of the previous generative models, such as GAN (Goodfellow et al., 2020). The rise of diffusion models (Song et al., 2020b), known for their generative capabilities, high sample diversity and inherent stochasticity, signals a promising direction for mitigating these persistent issues. DiffPure (Nie et al., 2022) proposed to use the diffusion model for adversarial purification. CLIPure (Zhang et al., 2025) operates directly in the CLIP latent space, correcting embeddings of adversarial examples for downstream tasks. However, previous works did not discuss the optimal number of steps for forward noise-injection. DiffPure (Nie et al., 2022) and CLIPure (Zhang et al., 2025) both use a fixed diffusion time, which is inflexible for adversarial inputs of diverse hardness. We propose a threshold-based stopping criterion in DiffCAP, and therefore reduces the number of noise-injection steps and improves performance by a significant margin.

## C  THEORETICAL PROOFS

### C.1  PROOF OF THM. 1

*Proof of Thm. 1.* Expanding the forward diffusion solution from Eq. (5), we get:

$$\boldsymbol{x}(t) = \sqrt{\alpha(t)}\,\boldsymbol{x}_{\text{adv}} + \sqrt{1-\alpha(t)}\,\boldsymbol{\epsilon} = \sqrt{\alpha(t)}\left[\boldsymbol{x} + \frac{\sqrt{1-a(t)}}{\sqrt{a(t)}}\boldsymbol{\epsilon} + \boldsymbol{\epsilon}_{\text{adv}}\right], \quad \boldsymbol{\epsilon} \sim \mathcal{N}(0, I).$$

Given Asm. 1 and the property of Gaussian distribution, we can analyze the classifier output as follows by absorbing the scaling factor $\sqrt{a(t)}$ and introduce $\boldsymbol{\epsilon}'$:

$$h(\boldsymbol{x}(t)) = h(\boldsymbol{x} + \boldsymbol{\epsilon}' + \boldsymbol{\epsilon}_{\text{adv}}), \quad \text{where } \boldsymbol{\epsilon}' \sim \mathcal{N}(0, \sigma(t)^2 I), \quad \sigma(t)^2 = \frac{1-\alpha(t)}{\alpha(t)}.$$

Applying the randomized smoothing bound from Theorem 1 of (Cohen et al., 2019), the classification is guaranteed to return class $k_1$ if:

$$\|\boldsymbol{\epsilon}_{\text{adv}}\|_2 < \frac{\sigma(t)}{2}\left(\Phi^{-1}(\underline{p_1}) - \Phi^{-1}(\overline{p_2})\right).$$

By re-organizing the term, we have:

$$\sigma(t) > \frac{2\|\boldsymbol{\epsilon}_{\text{adv}}\|_2}{\Phi^{-1}(\underline{p_1}) - \Phi^{-1}(\overline{p_2})}.$$

Since $\sigma(t)^2 = \frac{1-\alpha(t)}{\alpha(t)}$, this yields:

$$\frac{1-\alpha(t)}{\alpha(t)} > \left(\frac{2\|\boldsymbol{\epsilon}_{\text{adv}}\|_2}{\Phi^{-1}(\underline{p_1}) - \Phi^{-1}(\overline{p_2})}\right)^2.$$

By re-organizing the term, we obtain:

$$\alpha(t) < \frac{1}{1 + \left(\frac{2\|\boldsymbol{\epsilon}_{\text{adv}}\|_2}{\Phi^{-1}(\underline{p_1}) - \Phi^{-1}(\overline{p_2})}\right)^2}.$$

Since $\alpha(t) = \exp\left(-\int_0^t \beta(s)\,\mathrm{d}s\right)$, and with linear schedule $\beta(t) = \beta_{\min} + (\beta_{\max} - \beta_{\min})t$, we compute:

$$\int_0^t \beta(s)\mathrm{d}s = \beta_{\min}t + \frac{1}{2}(\beta_{\max} - \beta_{\min})t^2.$$

Thus,

$$\alpha(t) = \exp\left(-\beta_{\min}t - \frac{1}{2}(\beta_{\max} - \beta_{\min})t^2\right).$$

Let $M := \log\left(1 + \left(\frac{2\|\boldsymbol{\epsilon}_{\mathrm{adv}}\|_2}{\Phi^{-1}(\underline{p_1}) - \Phi^{-1}(\overline{p_2})}\right)^2\right)$. Then by setting:

$$\beta_{\min}t + \frac{1}{2}(\beta_{\max} - \beta_{\min})t^2 = M,$$

we obtain:

$$t = t_{\min} = \frac{2M}{\sqrt{\beta_{\min}^2 + 2(\beta_{\max} - \beta_{\min})M} + \beta_{\min}}.$$

Then, when $\beta_{\min} \geq M$, we have $t_{\min} \leq 1$, and for $t_{\min} \leq t \leq 1$, we have $\arg\max_{k \in [K]} \mathbb{P}(h(\boldsymbol{x}(t) = k) = k_1$. $\qquad\square$

## C.2 PROOF OF LEMMA 1

*Proof of Lemma 1.* we have:

$$\mathbb{E}\left[\|\phi(\boldsymbol{x}(t_1)) - \phi(\boldsymbol{x}(t_2))\|_2\right] \leq L \cdot \mathbb{E}\left[\|\boldsymbol{x}(t_1) - \boldsymbol{x}(t_2)\|_2\right] \tag{8}$$

$$\leq L \cdot \sqrt{\mathbb{E}\left[\|\boldsymbol{x}(t_1) - \boldsymbol{x}(t_2)\|_2^2\right]}, \tag{9}$$

where Eq. (8) follows from the Lipschitz continuity of $\phi$, Eq. (9) uses Jensen's inequality, i.e., $\mathbb{E}\left[\|Z\|\right] \leq \sqrt{\mathbb{E}\left[\|Z\|^2\right]}$ for any random vector $Z$.

Next, to compute $\mathbb{E}\left[\|\boldsymbol{x}(t_1) - \boldsymbol{x}(t_2)\|^2\right]$, use Eq. (5):

$$\boldsymbol{x}(t_1) - \boldsymbol{x}(t_2) = \left(\sqrt{\alpha(t_1)} - \sqrt{\alpha(t_2)}\right)\boldsymbol{x}_{\mathrm{adv}} + \left(\sqrt{1 - \alpha(t_1)} - \sqrt{1 - \alpha(t_2)}\right)\boldsymbol{\epsilon},$$

where $\boldsymbol{\epsilon} \sim \mathcal{N}(0, I)$. Taking the squared norm and expectation:

$$\mathbb{E}\left[\|\boldsymbol{x}(t_1) - \boldsymbol{x}(t_2)\|^2\right] = \left(\sqrt{\alpha(t_1)} - \sqrt{\alpha(t_2)}\right)^2\|\boldsymbol{x}_{\mathrm{adv}}\|^2 + \left(\sqrt{1 - \alpha(t_1)} - \sqrt{1 - \alpha(t_2)}\right)^2 \cdot \mathbb{E}\left[\|\boldsymbol{\epsilon}\|^2\right]$$

$$= \left(\sqrt{\alpha(t_1)} - \sqrt{\alpha(t_2)}\right)^2\|\boldsymbol{x}_{\mathrm{adv}}\|^2 + \left(\sqrt{1 - \alpha(t_1)} - \sqrt{1 - \alpha(t_2)}\right)^2 d.$$

As $t_1, t_2 \to 1$, both terms vanish, so the expectation tends to zero. $\qquad\square$

## C.3 PROOF OF THM. 2

Before proving Thm. 2, we need the following Lemma.

**Lemma 2.** *Let $\beta(t) = \beta_{\min} + (\beta_{\max} - \beta_{\min})t$ be a linear noise schedule with $\beta_{\min} > 0$ and $\beta_{\max} > \beta_{\min}$. Define:*

$$\alpha(t) := \exp\left(-\int_0^t \beta(s)\,\mathrm{d}s\right), \quad \text{and} \quad f(t) := \beta(t) \cdot \sqrt{\frac{\alpha(t)}{1 - \alpha(t)}}.$$

*Then $f(t)$ is strictly decreasing for all $t \in [0, 1)$.*

*Proof.* We first analyze the function $f(t)$ by taking its logarithm:

$$\log f(t) = \log \beta(t) + \frac{1}{2}\log\left(\frac{\alpha(t)}{1 - \alpha(t)}\right).$$

Differentiating and using the chain rule, we attain:

$$\frac{d}{dt}\log f(t) = \frac{\beta'(t)}{\beta(t)} + \frac{1}{2}\left(\frac{d}{dt}\log\alpha(t) - \frac{d}{dt}\log(1 - \alpha(t))\right)$$

$$= \frac{\beta'(t)}{\beta(t)} + \frac{1}{2} \cdot \alpha'(t)\left(\frac{1}{\alpha(t)} + \frac{1}{1 - \alpha(t)}\right) \tag{10}$$

$$= \frac{\beta'(t)}{\beta(t)} - \frac{1}{2} \cdot \frac{\beta(t)}{1 - \alpha(t)},$$

where we use $\alpha'(t) = -\beta(t) \cdot \alpha(t)$. Let's define $g(t) := \frac{d}{dt} \log f(t)$. We now analyze the sign of $g(t)$. First, observe that $\beta'(t) = \beta_{\max} - \beta_{\min} > 0$ is constant, $\beta(t) \geq \beta_{\min} > 0$, so $\frac{\beta'(t)}{\beta(t)}$ is strictly decreasing in $t$. Also $\alpha(t)$ is strictly decreasing, so $1 - \alpha(t)$ is strictly increasing, and hence $\frac{\beta(t)}{1-\alpha(t)}$ is strictly increasing. Therefore, $g(t)$ is strictly decreasing in $t$. To show that $g(t) < 0$ for all $t \in [0,1)$, it suffices to show $g(t) < 0$ near $t = 0$.

When $t \approx 0$, we take:

$$\beta(t) \approx \beta_{\min}, \quad \int_0^t \beta(s)\,\mathrm{d}s \approx \beta_{\min}t, \quad \alpha(t) = \exp(-\beta_{\min}t) \approx 1 - \beta_{\min}t + o(t).$$

Thus,

$$\frac{\beta(t)}{1 - \alpha(t)} \approx \frac{\beta_{\min}}{\beta_{\min}t} = \frac{1}{t}, \quad \text{which diverges as } t \to 0.$$

Given the result:

$$\frac{\beta'(t)}{\beta(t)} \approx \frac{\beta_{\max} - \beta_{\min}}{\beta_{\min}},$$

which is a finite number, thus, $g(t) \to -\infty$ as $t \to 0$. Since $g(t)$ is strictly decreasing, it follows that $g(t) < 0$ for all $t \in [0,1)$. This implies that $\log f(t)$ is strictly decreasing, and thus $f(t)$ is strictly decreasing as well. $\qquad\square$

Now we are ready to present the proof of Thm. 2.

*Proof of Thm. 2.* Let $\Delta(t, \delta) := \boldsymbol{x}(t + \delta) - \boldsymbol{x}(t)$. From the closed-form solution of the VP-SDE,

$$\boldsymbol{x}(t) = \sqrt{\alpha(t)}\boldsymbol{x}_{\mathrm{adv}} + \sqrt{1 - \alpha(t)}\boldsymbol{\epsilon},$$

we have:

$$\Delta(t, \delta) = \left(\sqrt{\alpha(t+\delta)} - \sqrt{\alpha(t)}\right)\boldsymbol{x}_{\mathrm{adv}} + \left(\sqrt{1 - \alpha(t+\delta)} - \sqrt{1 - \alpha(t)}\right)\boldsymbol{\epsilon}.$$

Let us define:

$$A := \sqrt{\alpha(t+\delta)} - \sqrt{\alpha(t)}, \quad B := \sqrt{1 - \alpha(t+\delta)} - \sqrt{1 - \alpha(t)}. \tag{11}$$

Then: $\Delta(t, \delta) = A\boldsymbol{x}_{\mathrm{adv}} + B\boldsymbol{\epsilon}$. By the Lipschitz property of $\phi$ and the Cauchy-Schwarz inequality:

$$\mathbb{E}\left[\|\phi(\boldsymbol{x}(t+\delta)) - \phi(\boldsymbol{x}(t))\|_2\right] \leq L \cdot \mathbb{E}\left[\|\Delta(t,\delta)\|_2\right] \leq L \cdot \sqrt{\mathbb{E}\left[\|\Delta(t,\delta)\|_2^2\right]}, \tag{12}$$

where $L \in \{L_1, L_2\}$. We now compute this second moment:

$$\mathbb{E}\left[\|\Delta(t,\delta)\|_2^2\right] = \mathbb{E}\left[\|A\boldsymbol{x}_{\mathrm{adv}} + B\boldsymbol{\epsilon}\|_2^2\right] = A^2\|\boldsymbol{x}_{\mathrm{adv}}\|_2^2 + B^2\mathbb{E}\left[\|\boldsymbol{\epsilon}\|_2^2\right].$$

Since $\boldsymbol{\epsilon}$ is standard Gaussian in $\mathbb{R}^d$, $\mathbb{E}[\|\boldsymbol{\epsilon}\|^2] = d$. Thus:

$$\mathbb{E}\left[\|\Delta(t,\delta)\|_2^2\right] = A^2\|\boldsymbol{x}_{\mathrm{adv}}\|_2^2 + B^2 d.$$

We now perform Taylor expansion for $A$ and $B$ with respect to $t$. First note that:

$$\frac{d}{dt}\alpha(t) = -\beta(t)\alpha(t), \quad \frac{d}{dt}\sqrt{\alpha(t)} = -\frac{\beta(t)}{2}\sqrt{\alpha(t)}.$$

Plugging the above equation back into Eq. (11), we derive:

$$A = \sqrt{\alpha(t+\delta)} - \sqrt{\alpha(t)} = -\frac{\beta(t)}{2}\sqrt{\alpha(t)}\delta + o(\delta).$$

Similarly:

$$\frac{d}{dt}\sqrt{1 - \alpha(t)} = \frac{\beta(t)\alpha(t)}{2\sqrt{1 - \alpha(t)}}.$$

Plugging the above equation back into Eq. (11):

$$B = \sqrt{1 - \alpha(t + \delta)} - \sqrt{1 - \alpha(t)} = \frac{\beta(t)\alpha(t)}{2\sqrt{1 - \alpha(t)}}\delta + o(\delta).$$

Take the square and sum them up:

$$A^2 + B^2 = \frac{\beta(t)^2\delta^2}{4}\left(\alpha(t) + \frac{\alpha(t)^2}{1 - \alpha(t)}\right) + o(\delta^2) = \frac{\beta(t)^2\alpha(t)\delta^2}{4(1 - \alpha(t))} + o(\delta^2).$$

Since $\|\boldsymbol{x}_{\text{adv}}\|^2 \leq d$, we gain:

$$\mathbb{E}\left[\|\Delta(t, \delta)\|_2^2\right] \leq \frac{d \cdot \beta(t)^2\alpha(t)}{4(1 - \alpha(t))}\delta^2 + o(\delta^2). \tag{13}$$

Plugging Eq. (13) into Eq. (12), we arrive:

$$\mathbb{E}\left[\|\phi(\boldsymbol{x}(t + \delta)) - \phi(\boldsymbol{x}(t))\|_2\right] \leq L \cdot \sqrt{\mathbb{E}\left[\|\Delta(t, \delta)\|_2^2\right]} = O\left(L \cdot \delta \cdot \beta(t)\sqrt{\frac{\alpha(t)}{1 - \alpha(t)}}\right).$$

Lastly, by Lemma 2, we prove that the bound on the right hand side decreases as $t$ grows. $\qquad\square$

## C.4 EXTENSION TO GENERAL VLM TASKS

Our theoretical analysis naturally extends to VLMs through two complementary perspectives:

- The classifier $h$ maps $\mathbb{R}^d \to [K]$. This naturally captures CLIP-based zero-shot classification, where the final prediction is a Softmax over cosine similarities between the image embedding and text prompts. For generative VLMs like LLaVA, this formulation holds under standard VQA setups. By adopting a structured prompting strategy (e.g., "*Classify this image into categories* $[1], ..., [K]$"), the mapping becomes a function from $x \in \mathbb{R}^d$ to a probability distribution over tokens representing the $K$ class indices. Thus, the theoretical guarantees for $h(x)$ directly apply to the VLM's decision-making process in discriminative tasks.

- Essentially, our theoretical contribution is not limited to the final output label but is rooted in the stability of the vision encoder's embedding space. VLMs generate text sequences conditioned on the image embedding $\phi(x)$. Thm. 2 proves the convergence and stability of this semantic embedding $\phi(x(t))$ during the diffusion process. Since the VLM's generation is a function of this embedding, establishing a certified recovery region serves as a necessary condition for robust generation, whether the downstream task is classification, captioning, or VQA.

# D ADDITIONAL EXPERIMENTS

## D.1 MORE IMPLEMENTATION DETAILS

For the forward diffusion, we schedule noise with $\beta_{\min} = 0.1$, $\beta_{\max} = 20$, and a fixed step size of 0.01. In the reverse generation, we employ guided diffusion with a step size of 0.015. For stronger attacks with $\epsilon > 4/255$, we set the minimum diffusion depth to 0.04 for sufficient denoising. All baseline methods are evaluated using their respective best-performing hyperparameters as reported in the original papers. Unless otherwise specified, the remaining setups of the experiments in Appx. E are the same as those in the main text.

To keep the evaluation of adaptive attacks on large VLMs computationally tractable, we randomly choose 100 images per dataset. BPDA (Athalye et al., 2018a) attacks are run for 50 iterations. For the forward pass, we execute $x' = \text{DiffCAP}(x)$ wrapped as a `torch.nn.Module` to serve as the first layer of the VLM. For the backward pass, we approximate the gradient of the DiffCAP($\cdot$) with respect to the input as the *identity matrix* ($\nabla_x\text{DiffCAP}(x) \approx I$), leveraging the "detach" trick in `PyTorch`. This implementation allows BPDA to fully exploit DiffCAP's knowledge, including the CLIP (ViT-B/32), the stopping rule, and the semantic similarity threshold. We integrate this BPDA module directly into the APGD framework. For EOT (Athalye et al., 2018b), we estimate the expected gradients by averaging the BPDA-derived gradients over three stochastic forward passes of the DiffCAP($\cdot$) for attack optimization. BPDA and EOT ensure the adaptive adversarial examples are generated with stable gradient flow and robust to the randomness from the diffusion sampling.

Table 5: Top-1 accuracy (%) of CLIP in ZSC task with clean images and adversarial perturbations of two sizes under different defenses. $2$ and $4$ with TeCoA and FARE suggest the version that is fine-tuned by $\ell_\infty^{2/255}$ and $\ell_\infty^{4/255}$ bounded adversarial examples, respectively. The best result is in **bold** and the runner-up is underlined.

| Defense | CalTech101 | | | ImageNet1K | | |
|---|---|---|---|---|---|---|
| | clean | $\ell_\infty^{2/255}$ | $\ell_\infty^{4/255}$ | clean | $\ell_\infty^{2/255}$ | $\ell_\infty^{4/255}$ |
| No defense | 83.3 | 0.0 | 0.0 | 87.9 | 0.0 | 0.0 |
| TeCoA[2] (Mao et al., 2022) | 80.7 | 70.2 | 57.4 | 80.1 | 58.8 | 36.7 |
| FARE[2] (Schlarmann et al., 2024) | **84.8** | 73.0 | 46.6 | 85.5 | 56.5 | 25.6 |
| TeCoA[4] (Mao et al., 2022) | 78.4 | 69.7 | 60.9 | 74.3 | 59.2 | 41.9 |
| FARE[4] (Schlarmann et al., 2024) | 84.7 | 76.7 | 64.1 | 80.2 | 61.6 | 40.6 |
| JPEG-DL (Salamah et al., 2024) | 83.9 | 68.5 | 33.4 | **87.8** | 48.5 | 16.4 |
| DiffPure (Nie et al., 2022) | 83.6 | **83.2** | **83.1** | 81.0 | 79.7 | 79.7 |
| CLIPure (Zhang et al., 2025) | 82.9 | 80.8 | 80.1 | 87.7 | **85.4** | **84.6** |
| DiffCAP | 82.6 | 81.7 | 80.9 | 87.2 | 84.4 | 81.1 |

Table 6: CIDEr scores with VLMs OF-9B and LLaVA 1.5-7B on datasets COCO and Flickr30k.

| Defense | OF-COCO | OF-Flickr30k | LLaVA-COCO | LLaVA-Flickr30k |
|---|---|---|---|---|
| Clean | 79.7 | 60.1 | 115.5 | 77.5 |
| No defense | 0.9 | 0.3 | 1.5 | 0.5 |
| JPEG-DL | 5.2 | 3.2 | 5.0 | 2.7 |
| DiffPure | 72.7 | 49.2 | 100.0 | 62.5 |
| DiffCAP | **74.8** | **53.2** | **109.7** | **67.6** |

## D.2 ZERO-SHOT CLASSIFICATION

From Tab. 5, we observe that JPEG-DL underperforms compared to TeCoA and FARE, particularly under stronger attacks. CLIPure recovers its defense effectiveness, as only the `[CLS]` token is involved in prediction and no generative decoding is required. On CalTech101, DiffCAP outperforms CLIPure under attacks, and on ImageNet1K, DiffCAP achieves higher robustness than DiffPure. These results confirm the generalizability of DiffCAP: it not only excels in complex vision-language tasks requiring rich semantics but also delivers stable performance on standard classification benchmarks with more sparse semantic demands.

## D.3 LARGER ATTACK BUDGET

We conducted additional experiments on the IC task using the larger budget of $\ell_\infty^{16/255}$. We excluded TeCoA, FARE, and CLIPure from this evaluation, since they exhibited limited performance even under smaller perturbations. The results are reported in Tab. 6.

Unlike under $\ell_\infty^{2/255}$ and $\ell_\infty^{4/255}$ attacks, JPEG-DL collapses with negligible improvement. While DiffPure demonstrates that diffusion-based purification remains viable at this magnitude, it consistently underperforms DiffCAP by a substantial margin across different settings.

## D.4 CLIP-AS-SURROGATE ATTACK ON MINIGPT 4-13B

We supply with experiments on MiniGPT 4-13B (Zhu et al., 2023). We evaluated DiffCAP with default hyperparameters against the AttackVLM (Zhao et al., 2023), which proposed more advanced transfer-based and query-based adversarial attacks, specifically designed to mislead VLMs into generating specific target captions, rather than plain untargeted degradation. We strictly follow their evaluation protocol and attack configurations.

Table 7: CLIP scores evaluating the DiffCAP against AttackVLM on MiniGPT 4-13B.

| Setting | RN50 | RN101 | ViT-B/32 | ViT-B/16 | ViT-L/14 | Avg. |
|---|---|---|---|---|---|---|
| Clean (Baseline) | 0.383 | 0.619 | 0.375 | 0.361 | 0.356 | 0.419 |
| MF-ii (No Defense) | 0.689 | 0.784 | 0.772 | 0.732 | 0.674 | 0.730 |
| MF-ii (DiffCAP) | **0.470** | **0.505** | **0.481** | **0.459** | **0.411** | **0.465** |
| MF-ii + MF-tt (No Defense) | 0.756 | 0.752 | 0.787 | 0.772 | 0.750 | 0.763 |
| MF-ii + MF-tt (DiffCAP) | **0.405** | **0.528** | **0.439** | **0.422** | **0.380** | **0.435** |

Table 8: Evaluation of DiffCAP in mitigating Hallucination and Jailbreaking of large VLMs.

| | Hallucination with LLaVA 1.5-13B | | | Jailbreaking with MiniGPT 4-13B | | | | |
|---|---|---|---|---|---|---|---|---|
| | adversarial | popular | random | any | identity | disinfo | crime | x-risk |
| Clean | 82.7 | 84.3 | 85.0 | 16/40 | 3/11 | 6/13 | 6/13 | 1/3 |
| Attack | N/A | N/A | N/A | 24/40 | 6/11 | 7/13 | 12/13 | 2/3 |
| DiffCAP | 83.2 | 85.1 | 86.3 | 14/40 | 3/11 | 5/13 | 5/13 | 1/3 |

The clean images are sampled from the ImageNet1K dataset and a target text is randomly selected from the COCO captions for each clean image. We report the CLIP score between the generated responses of input images and predefined targeted texts, as computed by various CLIP text encoders and their average. The prompt is fixed as "*what is the content of this image?*". Pretrained CLIP encoders (ViT-B/32) are used as *surrogate* models for attacks.

As shown in Tab. 7, the 'No Defense' settings yield high CLIP scores, indicating that the VLM was successfully manipulated into generating the target text. However, DiffCAP maintains its effectiveness under MF-ii (image-to-image transfer) and MF-ii + MF-tt (joint image-text query) attacks, reducing the CLIP scores comparable to, and in some cases lower than, the clean baseline. This verify the generalizability of DiffCAP for more complex VLM architectures and attack strategies.

## D.5 HALLUCINATION AND JAILBREAKING

Large VLMs tend to hallucinate objects that are not actually present in the image. POPE (Li et al., 2023) serves as a benchmark to formulate hallucination detection as a binary classification task. In Tab. 8, we report the F1-scores across three POPE categories using LLaVA 1.5-13B, with and without DiffCAP applied as image preprocessing. A consistent improvement is observed with DiffCAP. This suggests that DiffCAP, through Langevin dynamics, walks image features to semantically stable regions of the distribution. By suppressing high-frequency adversarial or spurious signals, DiffCAP becomes less sensitive to misleading cues and more robust against hallucination.

Large VLMs are also vulnerable to jailbreaking attacks on the visual modality (Carlini et al., 2024; Qi et al., 2024), where adversarially crafted images can induce harmful outputs in response to restricted prompts (e.g., "*How to make a bomb?*"). We apply the attack proposed by (Qi et al., 2024) to MiniGPT 4-13B (Zhu et al., 2023) and count the policy-violating outputs triggered by 40 harmful prompts spanning four categories. Even under a stronger perturbation budget ($\ell_\infty^{16/255}$), DiffCAP successfully restores the model's behavior to a level comparable to, or slightly better than, the clean condition. These findings reinforce the versatility of DiffCAP as a modular defense measure, readily adaptable to various VLMs and tasks requiring robustness guarantee. As jailbreaking attacks continue to evolve rapidly, benchmarking DiffCAP against such threats falls outside the scope of this work, but nonetheless marks a promising direction for future investigation.

## D.6 PERCEPTUAL QUALITY

We assessed the perceptual quality and semantic preservation of the purified images compare to their clean counterparts on the IC task under $\ell_\infty^{4/255}$ adversarial perturbations. We employed three standard metrics: PSNR (Peak Signal-to-Noise Ratio) measures pixel-level signal fidelity; LPIPS

Table 9: Perceptual quality and semantic preservation. ↑ for higher is better, and ↓ for lower is better.

| Setting | PSNR (dB) ↑ | | LPIPS ↓ | | CLIP Score ↑ | |
|---|---|---|---|---|---|---|
| | DiffPure | DiffCAP | DiffPure | DiffCAP | DiffPure | DiffCAP |
| OF-COCO | 24.27 | **29.03** | 0.296 | **0.160** | 0.871 | **0.930** |
| OF-Flickr30k | 23.44 | **28.21** | 0.298 | **0.151** | 0.865 | **0.930** |
| LLaVA-COCO | 25.41 | **29.76** | 0.304 | **0.200** | 0.849 | **0.893** |
| LLaVA-Flickr30k | 24.59 | **29.08** | 0.306 | **0.194** | 0.837 | **0.884** |

Table 10: VQA accuracy (%) on the TextVQA dataset with OF-9B under different attack radii.

| | clean | $\ell_\infty^{2/255}$ | $\ell_\infty^{4/255}$ | $\ell_\infty^{8/255}$ | $\ell_2^{2/255}$ | $\ell_2^{4/255}$ | $\ell_2^{8/255}$ |
|---|---|---|---|---|---|---|---|
| No defense | 23.8 | 0.0 | 0.0 | 0.0 | 6.2 | 6.0 | 4.8 |
| DiffCAP | 18.6 | 16.2 | 16.7 | 16.7 | 16.5 | 16.6 | 16.2 |

(Zhang et al., 2018) (Learned Perceptual Image Patch Similarity) measures perceptual distance using deep features, aligning closely with human visual perception; CLIP Score measures the semantic consistency.

The quantitative comparison between DiffCAP and DiffPure is presented in Tab. 9. DiffCAP significantly outperforms DiffPure across all three metrics with different VLMs and datasets. The higher PSNR and lower LPIPS indicate that DiffCAP preserves fine-grained visual details and reduces perceptual artifacts effectively. The superior CLIP Scores further validate our method's high semantic integrity, establishing a new SOTA balance between robustness and faithfulness among purification methods.

### D.7 OTHER DISCUSSION

$\ell_2$ **bounded threats.** We perform a test of the VQA task, where Tab. 10 compares DiffCAP with no-defense baseline. The results demonstrate that DiffCAP maintains its effectiveness regardless of whether the adversarial perturbations follow $\ell_\infty$ or $\ell_2$ norm bound.

**End-to-end runtime and memory comparisons.** We report the additional computational overhead introduced by the defense mechanisms, excluding the standard VLM inference portion. All measurements were performed with a batch size of 1, on the COCO with OF and $\ell_\infty^{2/255}$ adversarial examples.

For adversarially fine-tuned vision encoders (TeCoA and FARE), the computational burden is entirely from training phase. Once deployed, these methods incur zero additional inference latency and no extra memory overhead compared to the original VLMs, as they simply replace the weights of vision encoder. However, they require massive computational resources for training on perturbed data. For test-time defenses, we report the comparison of runtime and memory usage in the Tab. 11.

**Under- and Over-Purification.** We conduct a trial of IC task on the COCO dataset with LLaVA 1.5-7B. Tab. 12 records the reductions in CIDEr scores of DiffPure with two diffusion time compared to DiffCAP under three attack radii. DiffPure with a lower fixed diffusion time suffers from under-purification (insufficient robustness) against stronger attacks, while using a higher fixed diffusion time leads to over-purification (visual artifacts) against weaker attacks. This underscores the benefit of DiffCAP's design, where the dynamic diffusion mechanism determines the optimal purification extent for different attack vectors.

## E SYSTEMATIC ANALYSIS OF HYPERPARAMETERS

### E.1 THE VARIANCE OF THE THRESHOLD

Table 11: Runtime and memory comparison.

| Method | Time per Image | Peak GPU Memory |
|--------|----------------|-----------------|
| JPEG-DL | 0.01 s | N/A (CPU) |
| DiffPure | 2.3 s | 2.7 GB |
| CLIPure | 0.01 s | 0.3 GB |
| DiffCAP | 1.1 s | 2.5 GB |

Table 12: The CIDEr score gains of DiffCAP over DiffPure. $\ell_\infty^{8/255}$ attack is with BPDA.

| Diffusion time | $\ell_\infty^{2/255}$ | $\ell_\infty^{4/255}$ | $\ell_\infty^{8/255}$ |
|----------------|------------|------------|------------|
| 0.020 | 5.2 | 5.9 | 18.2 |
| 0.075 | 11.2 | 11.9 | 11.7 |

The number of images in the calibration set for Alg. 2 has a faint impact on threshold calculation. When we vary the subset size of image pairs to 100, 200, 300, and use three different random seeds, the resulting $\tau$ remains stable at $0.958 \pm 0.003$.

### E.2 THE THRESHOLD FOR OUT-OF-DISTRIBUTION (OOD)

Severe distribution shifts naturally degrade VLM performance compared to in-domain natural images. For example, VLMs can underperform simple Convolutional Neural Networks (CNNs) on datasets like MNIST (Deng, 2012). However, our empirical results demonstrate that $\tau$ is highly robust, and Alg. 2 serves as an optimizer when representative data (e.g., medical or satellite image) are available. The calibration provides a minor performance boost typical of hyperparameter tuning but is not required to achieve SOTA defense.

We first deliver a validation on the MNIST dataset for ZSC task with CLIP. The results in Tab. 13 suggest that DiffCAP maintains strong performance even when the domain departs from natural, real-world imagery that characterizes most VLM application scenarios. While the threshold $0.96$ is already robust for this non-natural, digital domain, using the threshold $0.97$ computed on the MNIST via Alg. 2 brings a slight gain in performance.

To further analyze how semantic complexity affects the acquisition of $\tau$, we conducted additional experiments on two datasets with distinct styles: ImageNet-R (Hendrycks et al., 2021) (art, cartoons, and other diverse styles representing *rich* semantics) and ImageNet-S (Wang et al., 2019) (black and white outlines representing *sparse* semantics). We compressed resolutions and crafted long-tail categories to simulate the extreme OOD conditions.

Alg. 2 calibrates $\tau = 0.95$ for ImageNet-R and $\tau = 0.97$ for ImageNet-S. This aligns with our intuition: rich semantics are easier to stabilize, necessitating a slightly lower threshold, whereas sparse semantics reach the recovery region harder. The former contains redundant textural and color cues that facilitate faster feature reconstruction by the diffusion model, and the latter is more sensitive to noise interference. Despite these differences, the performance variance across the $\tau \in [0.95, 0.97]$ interval is inapparent. Even using a sub-optimal threshold, DiffCAP consistently outperforms the DiffPure.

The results in Tab. 14 corroborated that $\tau$ is not brittle to severe domain shifts in terms of both measurement and deployment. Users can safely tune within the $0.95 \sim 0.97$ "safety belt" without the risk of losing SOTA performance, although Alg. 2 allows for a quicker hyperparameter search.

### E.3 THE THRESHOLD FOR DIFFERENT TASKS

The calculation of $\tau$ by Alg. 2 is based on the semantic stability of image embeddings and is therefore *task-agnostic*. Here we verify whether the calibrated threshold remains optimal across different downstream tasks. As shown in Fig. 2, for the IC task on COCO with OF-9B under $\ell_\infty^{2/255}$ attack, the

Table 13: Top-1 accuracy (%) on 1,000 randomly selected MNIST test images under $\ell_\infty^{4/255}$ attack.

| Clean | No defense | DiffCAP ($\tau = 0.96$) | DiffCAP ($\tau = 0.97$) |
|-------|-----------|------------------------|------------------------|
| 75.2 | 0.0 | 79.6 | 80.9 |

Table 14: Top-1 accuracy (%) of ZSC task with CLIP under $\ell_\infty^{4/255}$ attack on OOD datasets.

| Setting | ImageNet-R(endition) | ImageNet-S(ketch) |
|---------|----------------------|-------------------|
| Clean | 75.5 | 58.5 |
| No defense | 0.0 | 0.1 |
| DiffPure | 63.5 | 50.1 |
| DiffCAP ($\tau = 0.95$) | **69.2** | 51.9 |
| DiffCAP ($\tau = 0.96$) | 67.6 | 52.5 |
| DiffCAP ($\tau = 0.97$) | 66.7 | **53.1** |

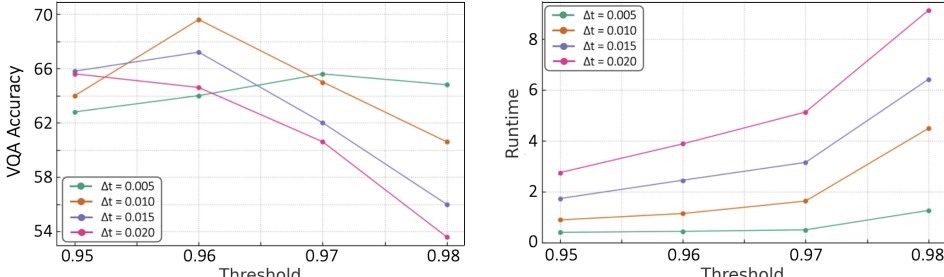

Figure 5: VQA Accuracy (%) and running time (in seconds) per image with varying thresholds $\tau$ and diffusion step sizes ($\Delta t$) for DiffCAP. The evaluation is based on the VQA task under $\ell_\infty^{4/255}$ attack.

calibrated $\tau = 0.96$ delivers the overall highest CIDEr scores across various diffusion step sizes $\Delta t$, reflecting the effectiveness of Alg. 2.

To demonstrate task transferability of $\tau$, we conducted an analogous ablation study for the VQA task on VQAv2 with LLaVA 1.5-7B under $\ell_\infty^{4/255}$ attack. The Fig. 5 visualizes the quantitative results, which indicate that $\tau = 0.96$ achieves the highest accuracy at the default step size $\Delta t = 0.010$ while maintaining the efficiency. This is consistent with our observation in Fig. 2.

### E.4 THE THRESHOLD FOR VARIOUS CLIP

Alg. 2 employs a vision encoder to quantify semantic stability. To assess whether the encoder architecture biases the calibration of $\tau$, we modifies Alg. 2 with several CLIP variants. The calibrated $\tau$ values are listed in Tab. 15, clustering around 0.96.

We posit that for natural images, unless the encoder architecture is fundamentally altered (e.g., deviating from the *contrastive learning* paradigm), the semantic threshold is primarily governed by the underlying data distribution rather than the specific vision backbone. This clue retrospectively explains the results in Tab. 4, where we revealed that $\tau = 0.96$ remains effective when the DiffCAP vision encoder is replaced by other CLIP variants. Overall, DiffCAP is largely insensitive to the vision backbone used for either calibration or purification.

### E.5 THE STEP SIZE AND ROBUSTNESS-FIDELITY

The trade-off is primarily influenced by the diffusion depth: deeper diffusion enhances robustness but risks losing information, while shallower diffusion preserves details but leaves residual adversarial

Table 15: The calibrated threshold across different vision backbones via Alg. 2.

|  | RN50 | RN101 | ViT-B/32 | ViT-B/16 | ViT-L/14 | Avg. |
|---|---|---|---|---|---|---|
| $\tau$ | 0.964 | 0.967 | 0.958 | 0.956 | 0.962 | **0.961** |

Table 16: Comparison of different scheduling strategies.

| Setting | Linear | Cosine | Exponential Decay |
|---|---|---|---|
| CIDEr Score for OF-COCO-$\ell_\infty^{2/255}$ | 79.3 | 79.2 | 79.2 |
| VQA Acc. (%) for LLaVA-VQAv2-$\ell_\infty^{4/255}$ | 68.5 | 68.0 | 68.1 |

noise. In DiffCAP, the diffusion depth is dynamically determined by the coupling between the $\Delta t$ and $\tau$.

Larger $\Delta t$ "jumps farther" in the embedding space between steps. If combined with a high $\tau$, i.e., a strict stability requirement, the forward diffusion may *overshoot* the recovery region, leading to redundant steps and over-purification. Smaller $\Delta t$ induces finer granularity. If combined with a low $\tau$, the stability condition can be triggered too early, causing *premature* stopping and under-purification.

Therefore, an optimal trade-off requires balancing $\Delta t$ and $\tau$, avoiding configurations where they are simultaneously too large or too small. Fig. 2 and Fig. 5 argue that, across different tasks, datasets, attack magnitudes, and VLMs, there exists a relatively safe range of $\Delta t$ selection. With $\tau = 0.96$ fixed, $\Delta t \in [0.05, 0.15]$ consistently outperforms DiffPure, where $\Delta t = 0.10$ is set as a reliable and efficient default.

### E.6 THE JUSTIFICATION OF SCHEDULER

We employ a linear noise schedule in DiffCAP for theoretical guarantee, alignment with diffusion dynamics, and empirical performance.

Our theoretical derivations are explicitly formulated under the linear precondition. Regardless of whether the noise injection follows a linear or non-linear schedule, the cumulative noise will eventually push the image into the certified recovery region. With a sufficiently small step size ($\Delta t \approx 0.01$), the specific noise trajectory only marginally shifts the precise timestamp at which this region is entered but does not alter the fundamental semantic convergence behavior.

DiffCAP operates in *conjunction* with a pre-trained diffusion model for the reverse denoising step, where linear $\beta$ schedule is adopted. To experimentally support DiffCAP is invariant to the moderate noise schedule variations, we compare our default Linear scheduler against a Cosine ($\Delta t' = \Delta t \cdot \cos(\frac{\pi i}{2N})$) and an Exponential Decay ($\Delta t' = \Delta t \cdot 0.9^i$) scheduler, where $i$ denotes the step index and $N$ refers to the total number of steps. As presented in Tab. 16, the alternative schedulers exhibit no material performance differences.

