# OpenReview forum: "Diffusion-based Cumulative Adversarial Purification for Vision Language Models"
_ICLR.cc/2026/Conference — Submitted to ICLR 2026_

### Official Review · Reviewer_s6Go · 2025-10-26

**Soundness:** 3
**Presentation:** 2
**Contribution:** 2
**Rating:** 6
**Confidence:** 3

**Summary:**

This paper introduces **DiffCAP**, a diffusion-based cumulative adversarial purification framework for vision-language models (VLMs). It adaptively determines the minimal diffusion steps by monitoring the cosine similarity between semantic embeddings, thus improving efficiency while maintaining robustness. The authors provide theoretical guarantees for convergence and demonstrate strong empirical results across multiple datasets and models.

**Strengths:**

* Theoretical proofs guarantee the convergence of semantic embeddings under forward diffusion and define a region where adversarial perturbations vanish.
* Extensive experiments validate the effectiveness of the proposed method.
* Unlike DiffPure’s fixed diffusion length, DiffCAP dynamically determines the minimal diffusion time via cosine similarity of VLM embeddings — an elegant and effective per-input adaptivity.

**Weaknesses:**

* The method represents an incremental improvement over DiffPure.
* Although diffusion steps are adaptively determined per image, the use of a fixed similarity threshold may be domain-dependent and needs further validation.
* The implementation of adaptive attacks is not clearly explained; gradient explosion could occur when performing full differentiable attacks and should be clarified.
* Using CLIP-B/32 for similarity measurement may be unreliable if attackers exploit the same encoder to craft adversarial examples.

**Questions:**

1. Clarify how adaptive white-box attacks are implemented and whether gradient instability was observed.
2. Analyze the sensitivity of the similarity threshold τ across different domains and models.
3. Discuss potential vulnerabilities when using CLIP-B/32 as the embedding backbone, and evaluate robustness under CLIP-aware adaptive attacks.
4. Provide additional justification to highlight conceptual differences from DiffPure and demonstrate that DiffCAP is more than an incremental improvement.

---

> ### Author Response · Authors · 2025-11-22
>
> We sincerely thank the reviewer for their time and constructive feedback, which has helped us improve the quality of our work. In this response, we have addressed all weaknesses/questions **unrelated to the hyperparameter settings**. Regarding the inquiries specifically concerning the threshold, we are currently conducting experiments to provide a systematic analysis. We plan to post a unified response covering all $\tau$-related concerns by **Nov. 29**, and we greatly appreciate your patience meantime. Following ICLR’s suggestion, we are posting these initial responses to ensure thorough rebuttal rather than rushing in the final days. We guarantee that all new results, clarifications, and revisions discussed here will be fully integrated into the updated manuscript by **Dec. 3**.
>
> ---
>
> **Answer to W1 and Q4:**
>
> DiffCAP drives DiffPure from the **static, empirical purification logic to the dynamic, theoretically grounded semantic restoration** with three key innovations:
>
> - DiffPure operates as a heuristic without proofs regarding the optimal diffusion timestep. In contrast, DiffCAP provides a rigorous theoretical framework. We establish a **certified recovery region** (Theorem 1) and quantify the **convergence rate of semantic variation** (Theorem 2). These theorems provide critical insights into the dynamics of VLM embedding stability under noise, extending beyond this specific method to the broader research problem of **safety alignment in the multimodal LLMs community**.
> - We are the **first** to propose a purification strategy that **determines the defense strength according to the attack difficulty of individual input**. DiffPure relies on a fixed diffusion time, leading to inevitable **under-purification** (inadequate robustness) or **over-purification** (information loss). DiffCAP’s semantic stopping criterion not only overcomes this dilemma, its instance-level adaptivity also introduces **inherent uncertainty and further hardness to design corresponding white-box attacks**.
> - DiffCAP consistently outperforms DiffPure and other selected defense baselines published within past two years' top-tier AI conferences by a **significant margin** across diverse tasks, VLMs, datasets, and attack configurations. Furthermore, DiffCAP **substantially reduces computational overhead** (reducing diffusion steps by $\sim 2/3$ compare to DiffPure), addressing the core efficiency bottleneck that previously hindered the deployment of diffusion-based defenses.
>
> In summary, DiffCAP is not an incremental improvement over DiffPure. It is a **certified, adaptive, and efficient** framework that redefines how generative purification should be deployed to modern VLMs.
>
> ---
>
> **Answer to W3 and Q1:**
>
> The diffusion process iteratively solves stochastic differential equations (SDEs) with hundreds of steps. Directly backpropagating through this chain creates a deep recursive computation graph that can cause vanishing or exploding gradients. On the other hand, the adaptive mechanism in DiffCAP based on semantic threshold introduces non-differentiable discrete logic. We addressed these two issues through the **Backward Pass Differentiable Approximation** (BPDA).
>
> For the forward pass, we execute $x_{in}' = \text{DiffCAP}(x_{in})$ wrapped as a torch.nn.Module to serve as the first layer of the downstream VLM. For the backward pass, we approximate the gradient of the $\text{DiffCAP}(\cdot)$ with respect to the input as the **identity matrix** ($\nabla_x \text{DiffCAP}(x) \approx I$), leveraging the "detach" trick in PyTorch.
> ```
> def diffcap_bpda(x: torch.Tensor, DiffCAP) -> torch.Tensor:
>     with torch.no_grad():
>         y = DiffCAP(x).to(x.dtype)  # Forward: DiffCAP(x) → y
>     return x + (y - x).detach()     # Backward: dy/dx → I
> ```
> This implementation ensures that during the optimization of the adversarial perturbation, the gradient flows correctly through the VLM ($\nabla_{x_{in}} \mathcal{L}$) and passes transparently through the $\text{DiffCAP}(\cdot)$ to the input $x_{in}$. **Although the gradient instability is not observed, the computation is inextricably expensive for large VLMs.** Thus, we randomly choose $100$ images per dataset for evaluation and run BPDA attack for $50$ iterations with an $\ell_{\infty}^{8/255}$ bound.
>
> For **Expectation over Transformation** (EOT), which we combine with BPDA to handle DiffCAP's stochasticity, we estimate the expected gradients by **averaging the BPDA-derived gradients over multiple stochastic forward passes** of the $\text{DiffCAP}(\cdot)$ for attack optimization (we take three in our experiments). This ensures the generated adaptive adversarial examples are robust to the randomness from the diffusion sampling process.

---

> ### Author Response · Authors · 2025-11-22
>
> **Answer to W4 and Q3:**
>
> As explained in our response to **W3/Q1**, the BPDA attack has the full knowledge of DiffCAP, including the CLIP (ViT-B/32) and the semantic similarity threshold. We also test DiffCAP against the **AttackVLM** (Zhao et al., 2023), which proposed more advanced **transfer-based** and **query-based adversarial attacks** specifically designed to mislead VLMs into generating specific target captions. Pretrained CLIP encoders (ViT-B/32) are used as surrogate models for attacks.
>
> We strictly follow their evaluation protocol on the **MiniGPT-4**, where the clean images are sampled from the ImageNet1K dataset and a target text is randomly selected from the COCO captions for each clean image. We report the **CLIP score** between the generated responses of input images and predefined targeted texts, as computed by various CLIP text encoders and their average. The prompt is fixed as “what is the content of this image?”.
>
> |  | RN50 | RN101 | ViT-B/32 | ViT-B/16 | ViT-L/14 | Ensemble |
> | --- | --- | --- | --- | --- | --- | --- |
> | clean (Baseline)   | 0.383 | 0.619 | 0.375 | 0.361 | 0.356 | 0.419 |
> | MF-ii (No Defense)   | 0.689 | 0.784 | 0.772 | 0.732 | 0.674 | 0.730 |
> | MF-ii (DiffCAP)  | **0.470** | **0.505** | **0.481** | **0.459** | **0.411** | **0.465** |
> | MF-ii + MF-tt (No Defense)   | 0.756 | 0.752 | 0.787 | 0.772 | 0.750 | 0.763 |
> | MF-ii + MF-tt (DiffCAP)   | **0.405** | **0.528** | **0.439** | **0.422** | **0.380** | **0.435** |
>
> As shown in the table, the 'No Defense' settings yield high CLIP scores, indicating that the VLM was successfully manipulated into generating the target text. However, **DiffCAP maintains its effectiveness** under MF-ii (image-to-image transfer) and MF-ii + MF-tt (joint image-text query) attacks, reducing the CLIP scores comparable to, and in some cases lower than, the clean baseline. The results verify the **generalizability of DiffCAP for CLIP-aware attack strategies**.

---

### Official Review · Reviewer_gzhU · 2025-10-31

**Soundness:** 3
**Presentation:** 3
**Contribution:** 3
**Rating:** 4
**Confidence:** 4

**Summary:**

This paper presents DiffCAP, a diffusion-based strategy that effectively neutralizes adversarial damage in VLMs. By adding minimal noise to adversarially damaged images, DiffCAP alters their latent embeddings within the VLM. The process involves progressively injecting random Gaussian noise until the embeddings of two consecutive noisy images reach a predefined similarity threshold, neutralizing the adversarial effect. A pre-trained diffusion model is then used to denoise the stabilized image, restoring a clear representation for VLM output generation. DiffCAP reduces hyperparameter tuning complexity and diffusion time, accelerating the denoising process. With strong theoretical and empirical support, DiffCAP provides a robust and practical solution for deploying VLMs securely in adversarial environments.

**Strengths:**

- DiffCAP proposed in the paper significantly outperforms existing defense techniques, providing a robust and practical solution for securely deploying VLMs in adversarial environments.
- The theoretical analysis presented in the paper provides a strong theoretical foundation for the DiffCAP approach.
- The paper is well-written, well-structured, and easy to follow.

**Weaknesses:**

- The paper proposes a diffusion-based adversarial purification method. Compared to previous diffusion-model-based purification approaches, the main improvement is the introduction of a learnable, adaptive forward-noise schedule. While the paper provides a relatively detailed theoretical analysis for this approach, the analysis is based on the assumption of a classifier. Extending it to vision-language models (VLMs) would require further explanation and justification.
- The paper lacks defense experiments under large adversarial perturbations, such as an attack budget of 16/255.
- Additionally, it is unclear whether DiffCAP is equally effective on other VLMs, such as Qwen-3VL, LLaVA-OneVision, MiniGPT, or InternVL-3.5.

**Questions:**

- The paper mentions that the threshold $\tau=0.96$ was determined using subsets of 100 randomly selected images from each of the datasets mentioned above. How sensitive is the threshold $\tau$ to the number of images in these subsets? Have experiments been conducted with different subset sizes? Furthermore, would using Algorithm 2 to determine separate thresholds $\tau$ for different datasets or tasks improve performance? Similarly, does the threshold vary across different CLIP architectures, and have experiments been conducted in this regard?
- For Figure 2, it is recommended to include the corresponding legends for the $x$- and $y$-axes directly on the figure for clarity.

---

> ### Author Response · Authors · 2025-11-22
>
> We sincerely thank the reviewer for their time and constructive feedback, which has helped us improve the quality of our work. In this response, we have addressed all weaknesses/questions **unrelated to the hyperparameter settings**. Regarding the inquiries specifically concerning the threshold, we are currently conducting experiments to provide a systematic analysis. We plan to post a unified response covering all $\tau$-related concerns by **Nov. 29**, and we greatly appreciate your patience meantime. Following ICLR’s suggestion, we are posting these initial responses to ensure thorough rebuttal rather than rushing in the final days. We guarantee that all new results, clarifications, and revisions discussed here will be fully integrated into the updated manuscript by **Dec. 3**.

---

> ### Author Response · Authors · 2025-11-22
>
> **Answer to W1:**
>
> Our analysis naturally extends to VLMs through two complementary perspectives:
>
> - The classifier $h$ maps $\mathbb{R}^d \to [K]$ (Line 202). This naturally captures CLIP-based zero-shot classification, where the final prediction is a **softmax over cosine similarities between the image embedding and text prompts**. For generative VLMs like LLaVA, this formulation holds under standard visual question answering (VQA) setups. By adopting a structured prompting strategy (e.g., "Classify this image into categories $[1],...,[K]$"), the **mapping** becomes a function from $x \in \mathbb{R}^d$ to a **probability distribution over tokens** representing the $K$ class indices. Thus, the theoretical guarantees for $h(x)$ directly apply to the VLM’s decision-making process in discriminative tasks.
> - More importantly, our theoretical contribution is not limited to the final output label but is rooted in the **stability of the vision encoder's embedding space**. VLMs generate text sequences conditioned on the image embedding $\phi(x)$. Theorem 2 proves the convergence and stability of this semantic embedding $\phi(x(t))$ during the diffusion process. Since the **VLM's generation is a function of this embedding**, establishing a certified recovery region serves as a necessary condition for robust generation, whether the downstream task is classification, captioning, or VQA.
>
> We will add a remark in Section 3 to explicitly bridge this connection between the classifier assumption and the VLM generation.
>
> ---
>
> **Answer to W2:**
>
> We conducted additional experiments on the Image Captioning task using requested budget of $\ell_{\infty}^{16/255}$. We excluded TeCoA, FARE, and CLIPure from this evaluation, since they exhibited limited performance even under smaller perturbations. The CIDEr scores are reported in the table below:
>
> |  | OF-COCO | OF-Flicker30k | LLaVA-COCO | LLaVA-Flicker30k |
> | --- | --- | --- | --- | --- |
> | Clean | 79.7 | 60.1 | 115.5 | 77.5 |
> | No defense | 0.9 | 0.3 | 1.5 | 0.5 |
> | JPEG-DL | 5.2 | 3.2 | 5.0 | 2.7 |
> | DiffPure | 72.7 | 49.2 | 100.0 | 62.5 |
> | DiffCAP | **74.8** | **53.2** | **109.7** | **67.6** |
>
> Unlike under $\ell_{\infty}^{2/255}$ and $\ell_{\infty}^{4/255}$ attacks, JPEG-DL collapses with negligible improvement. While DiffPure demonstrates that diffusion-based purification remains viable at this magnitude, it consistently underperforms DiffCAP by a substantial margin across different VLMs and datasets.
>
> ---
>
> **Answer to W3:**
>
> In this rebuttal, we supply with experiments on **MiniGPT-4**. We evaluated DiffCAP with default hyperparameters against the **AttackVLM** (Zhao et al., 2023), which proposed more advanced **transfer-based** and **query-based adversarial attacks**, specifically designed to mislead VLMs into generating specific target captions, rather than plain untargeted degradation. We strictly follow their evaluation protocol and attack configurations.
>
> The clean images are sampled from the ImageNet1K dataset and a target text is randomly selected from the COCO captions for each clean image. We report the **CLIP score** between the generated responses of input images and predefined targeted texts, as computed by various CLIP text encoders and their average. The prompt is fixed as “what is the content of this image?”. Pretrained CLIP encoders (ViT-B/32) are used as surrogate models for attacks.
>
> |  | RN50 | RN101 | ViT-B/32 | ViT-B/16 | ViT-L/14 | Ensemble |
> | --- | --- | --- | --- | --- | --- | --- |
> | clean (Baseline)   | 0.383 | 0.619 | 0.375 | 0.361 | 0.356 | 0.419 |
> | MF-ii (No Defense)   | 0.689 | 0.784 | 0.772 | 0.732 | 0.674 | 0.730 |
> | MF-ii (DiffCAP)  | **0.470** | **0.505** | **0.481** | **0.459** | **0.411** | **0.465** |
> | MF-ii + MF-tt (No Defense)   | 0.756 | 0.752 | 0.787 | 0.772 | 0.750 | 0.763 |
> | MF-ii + MF-tt (DiffCAP)   | **0.405** | **0.528** | **0.439** | **0.422** | **0.380** | **0.435** |
>
> As shown in above table, the 'No Defense' settings yield high CLIP scores, indicating that the VLM was successfully manipulated into generating the target text. However, **DiffCAP maintains its effectiveness** under MF-ii (image-to-image transfer) and MF-ii + MF-tt (joint image-text query) attacks, reducing the CLIP scores comparable to, and in some cases lower than, the clean baseline.
>
> Also in Appendix D.3 and Table 6, we utilized **MiniGPT-4** to evaluate DiffCAP’s effectiveness against **jailbreaking attacks**. We demonstrated that DiffCAP successfully restores the model’s safety alignment under strong adversarial perturbations ($\ell_{\infty}^{16/255}$). They together verify the **generalizability of DiffCAP for more complex VLM architectures and attack strategies**.
>
> ---
>
> **Answer to Q2:**
>
> We will update Figure 2 in the camera-ready version to explicitly include the axis labels (Threshold $\tau$ for the x-axis, and CIDEr Score/Runtime for the y-axis) to ensure clarity.

---

### Official Review · Reviewer_FdbB · 2025-10-31

**Soundness:** 4
**Presentation:** 4
**Contribution:** 2
**Rating:** 4
**Confidence:** 4

**Summary:**

This paper proposes DiffCAP, a diffusion-based cumulative adversarial purification method for VLMs. The key idea is to inject forward diffusion noise until the cosine similarity between consecutive VLM embeddings exceeds a threshold τ, then apply a pretrained diffusion model to reverse-denoise the stabilized image. The authors provide theory (a certified recovery region and a bound on semantic change across forward steps), and report improvements over adversarially trained encoders and purification baselines (e.g., DiffPure, CLIPure) on IC/VQA/ZSC across multiple datasets and attack strengths.

**Strengths:**

- Clear motivation vs. fixed-time purification: The paper pinpoints a limitation of prior works (DiffPure/CLIPure) that rely on a fixed diffusion time and proposes an input-adaptive stopping rule.
- Theoretical grounding: Thm. 1 formalizes a certified recovery region; Thm. 2 quantifies how the embedding change between adjacent forward steps shrinks toward terminal time (under a linear schedule).
- Simple, plug-in pipeline: DiffCAP requires no retraining of the downstream VLM and can wrap existing models.

**Weaknesses:**

- Adaptive steps vs. timesteps/step size analysis is limited; τ estimation may not generalize: Although the paper contrasts fixed-time baselines and includes an ablation over thresholds and step sizes (Fig. 2), it does not systematically quantify how different timestep schedules, step sizes (Δt), or stopping times impact robustness–fidelity across tasks/models/attack budgets. The τ selection uses mean similarity from a clean–adversarial pair set (Alg. 2); while a brief MNIST check is given, a broader out-of-domain analysis (style, resolution, long-tail categories) is missing. This raises questions about generalization and robustness when protected images lie outside the calibration domain.
- Computational overhead is under-characterized relative to baselines: The paper presents a diffusion-time box plot and notes a practical Δt trade-off, but lacks a unified, end-to-end comparison against all baselines under the same hardware/batch settings, plus a breakdown of time share (noise injection vs. embedding passes vs. reverse denoising), memory footprint, and τ-calibration cost.
- Threat-model scope (gray-box) leaves white-box adaptivity underexplored: Attacks assume gradient access to the task model but not the defense pipeline; while BPDA/EOT are considered, a fully white-box attacker aware of the stopping rule/τ and differentiating through the similarity check is not reported in the main text.
- Calibration details and reproducibility: τ is fixed to 0.96 from 100 pairs; the paper should report variance over random seeds/pair subsets and whether per-dataset or per-task τ materially helps (or hurts) clean fidelity.

**Questions:**

- The threshold τ is calibrated on a fixed set of clean–adversarial pairs. How well does it generalize to out-of-distribution data? Have you explored task- or dataset-specific τ values?
- Can you provide end-to-end runtime and memory comparisons with all baselines under identical hardware and batching conditions? Specifically, what is the time breakdown across noise injection, embedding computation, and reverse denoising?
- The current evaluation assumes the attacker cannot differentiate through the stopping rule. How does DiffCAP perform against a fully white-box adversary who knows τ and can approximate gradients through the cosine similarity check (e.g., via differentiable surrogates)? - How do different diffusion step sizes (Δt) or non-linear time schedules affect the robustness–fidelity trade-off? Is the current linear schedule and fixed Δt sufficiently justified across diverse models and attack budgets?
- Which pretrained diffusion model and scheduler do you use？

---

> ### Author Response · Authors · 2025-11-22
>
> We sincerely thank the reviewer for their time and constructive feedback, which has helped us improve the quality of our work. In this response, we have addressed all weaknesses/questions **unrelated to the hyperparameter settings**. Regarding the inquiries specifically concerning the threshold, we are currently conducting experiments to provide a systematic analysis. We plan to post a unified response covering all $\tau$-related concerns by **Nov. 29**, and we greatly appreciate your patience meantime. Following ICLR’s suggestion, we are posting these initial responses to ensure thorough rebuttal rather than rushing in the final days. We guarantee that all new results, clarifications, and revisions discussed here will be fully integrated into the updated manuscript by **Dec. 3**.
>
> ---
>
> **Answer to W2 and Q2:**
>
> We report the additional computational overhead introduced by the defense mechanisms, **excluding the standard VLM inference portion**. All measurements were performed on a single NVIDIA A100 (40G) GPU with a batch size of $1$, on the OF-COCO setting and $\ell_{\infty}^{2/255}$ adversarial examples.
>
> For adversarially fine-tuned vision encoders (TeCoA and FARE), the computational burden is entirely from training phase. Once deployed, these methods incur zero additional inference latency and no extra memory overhead compared to the original VLMs, as they simply replace the weights of vision encoder. However, **they require massive computational resources for training on perturbed data**, and we cannot find the description of respective time and memory consumption in their original papers.
>
> | |Time per Image|Peak GPU Memory|
> | -------- | -------- | -------- |
> | JPEG-DL | 0.01 s | N/A (CPU)|
> | DiffPure  | 2.3 s    | 2.7 GB   |
> | CLIPure  | 0.01 s  | 0.3 GB   |
> | DiffCAP  | 1.1 s    | 2.5 GB   |
>
> For test-time defenses, we report the comparison of runtime and memory usage in the table above. The time breakdown for DiffCAP (Line 470-478) is as follows: i) The one-time threshold $\tau$ calculation takes $\sim 3.4$ seconds in total for the calibration set. ii) The embedding extraction consumes only $\sim 6$ milliseconds per iteration and the Gaussian noise injection costs $\sim 4$ milliseconds per iteration. iii) Reverse denoising dominates the remaining runtime.
>
> ---
>
> **Answer to W3 and Q3:**
>
> The diffusion process iteratively solves stochastic differential equations (SDEs) with hundreds of steps. **Directly backpropagating through this chain creates a deep recursive computation graph that can cause vanishing or exploding gradients.** On the other hand, the adaptive mechanism in DiffCAP based on semantic threshold introduces non-differentiable discrete logic. We addressed these two issues through the Backward Pass Differentiable Approximation (BPDA).
>
> For the forward pass, we execute $x_{in}' = \text{DiffCAP}(x_{in})$ wrapped as a torch.nn.Module to serve as the first layer of the downstream VLM. For the backward pass, we approximate the gradient of the $\text{DiffCAP}(\cdot)$ with respect to the input as the **identity matrix** ($\nabla_x \text{DiffCAP}(x) \approx I$), leveraging the "detach" trick in PyTorch.
> ```
> def diffcap_bpda(x: torch.Tensor, DiffCAP) -> torch.Tensor:
>     with torch.no_grad():
>         y = DiffCAP(x).to(x.dtype)  # Forward: DiffCAP(x) → y
>     return x + (y - x).detach()     # Backward: dy/dx → I
> ```
> This implementation represents a **fully white-box** scenario as the BPDA has the whole knowledge of the DiffCAP, including the **CLIP (ViT-B/32)**, the **stopping rule**, and the **semantic similarity threshold**. The attack exploits the exact gradients of the victim VLM rather than a surrogate. We integrate this BPDA module directly into the Auto-PGD framework, **maximizing the loss on the purified output**.
>
> We perform a cross-check to verify whether the awareness of the specific DiffCAP hyperparameters can affect the success of the BPDA attack. We randomly choose $100$ images from COCO and run BPDA attack for $50$ iterations. When the attacker with perturbation budget of $\ell_{\infty}^{8/255}$ assuming $\tau=0.96$, the CIDEr score by OF decreases from $90.1$ (clean) to $27.1$.  When DiffCAP is deployed with $\tau=0.96$, it recovers the score to $79.9$, and increasing the threshold to $\tau = 0.97$ further improves performance to $82.1$. We can confirm that the adaptive attacker indeed exploits the DiffCAP mechanism when crafting adversarial examples.
>
> ---
>
> **Answer to Q4:**
>
> We leverage the OpenAI guided-diffusion model pretrained on the unconditional ImageNet 256x256 (Line 322). The noise scheduler is linear with $\beta_{min} = 0.1$ and $\beta_{max} = 20$. The step size for the forward diffusion and the reverse sampling are $0.01$ and $0.015$, respectively (Line 1011).

---

### Official Review · Reviewer_b8i8 · 2025-11-02

**Soundness:** 3
**Presentation:** 4
**Contribution:** 3
**Rating:** 6
**Confidence:** 4

**Summary:**

This paper introduces DiffCAP, a novel diffusion-based adversarial purification method for VLMs. The key innovation is a cumulative noise injection approach with an adaptive stopping criterion based on VLM embedding similarity, which dynamically determines the minimal diffusion steps needed for purification. The method is supported by theoretical analysis establishing a certified recovery region and quantifying semantic convergence rates during forward diffusion. Experiments across three VLMs and six datasets demonstrate superior performance compared to existing defenses.

**Strengths:**

* The paper provides rigorous theoretical contributions with Theorem 1 (certified recovery region) and Theorem 2 (convergence rate quantification). The theoretical analysis effectively motivates the algorithm design and provides insights into why cumulative diffusion works.
* The per-input adaptive diffusion time is a significant improvement over fixed-time approaches like DiffPure. This addresses a key limitation in prior work and reduces computational overhead while maintaining effectiveness.
* The method is shown to maintain strong robustness even against adaptive white-box attacks like BPDA and BPDA+EOT with a high attack budget ($l_{\infty}^{8/255}$), which is crucial for a credible defense mechanism.

**Weaknesses:**

* The paper only tested two large VLMs (OpenFlamingo and LLaVA-1.5). Why were only these two models tested? Are they representative?
* There is little discussion of the potential trade-offs between robustness and semantic fidelity during the denoising process. Adversarial purification methods like DiffCAP aim to improve robustness, but they might introduce distortions or loss of information in the process, especially when dealing with complex multimodal data.
* The method's key hyperparameter, the similarity threshold $\tau$, is determined via a separate calibration Algorithm 2 on a subset of data. While the paper justifies the re-usability of $\tau=0.96$ for natural images and shows it's robust to domain shift, this preliminary calibration step is still necessary, introducing a dependency on a clean, representative dataset for optimal performance in a new domain.

**Questions:**

Please refer to the weakness part.
And some suggestions:
* Test the defense on models beyond the chosen VLMs, such as newer or more complex architectures, to ensure that DiffCAP can generalize to a wider variety of models and attack scenarios.
* Evaluate perceptual quality and semantic preservation of DiffCAP’s purified outputs, particularly in image generation or image captioning tasks, to ensure that robust defenses do not come at the cost of severely distorting the input data.

---

> ### Author Response · Authors · 2025-11-22
>
> We sincerely thank the reviewer for their time and constructive feedback, which has helped us improve the quality of our work. In this response, we have addressed all weaknesses/questions **unrelated to the hyperparameter settings**. Regarding the inquiries specifically concerning the threshold, we are currently conducting experiments to provide a systematic analysis. We plan to post a unified response covering all $\tau$-related concerns by **Nov. 29**, and we greatly appreciate your patience meantime. Following ICLR’s suggestion, we are posting these initial responses to ensure thorough rebuttal rather than rushing in the final days. We guarantee that all new results, clarifications, and revisions discussed here will be fully integrated into the updated manuscript by **Dec. 3**.
>
> ---
>
> **Answer to W1:**
>
> **Representativeness of OpenFlamingo (OF) and LLaVA-1.5.** We selected OF and LLaVA because they represent the two dominant paradigms in open-source multimodal LLM architectures. OF represents the **interleaved pre-training paradigm**, where the model is trained on interleaved image and text sequences. It is a standard benchmark for few-shot and in-context learning capabilities. LLaVA is currently the most popular baseline for **visual instruction tuning paradigm**, which connects a pre-trained vision encoder to an LLM via a projector and is fine-tuned on instruction-following data. By evaluating these two, we cover the distinct architectural styles currently widely adopted in the community.
>
> **Evaluation on Additional Models.** Beyond the main experiments on OF and LLaVA, we evaluated other models and scales in our paper, which may have been overlooked. In Appendix D.3, we utilized **MiniGPT-4** to evaluate DiffCAP’s effectiveness against **jailbreaking attacks**. We demonstrated that DiffCAP successfully restores the model’s safety alignment under strong adversarial perturbations ($\ell_{\infty}^{16/255}$). Also in Appendix D.3, we employed the **larger 13B parameter version of LLaVA** to demonstrate scalability for the **hallucination mitigation** task, showing consistent performance improvements. In Appendix D.2, we evaluated **CLIP** directly for the **Zero-Shot Classification** task.
>
> **Model-Agnostic Nature of DiffCAP.** DiffCAP is an input-level purification strategy that removes adversarial noise from the image before it is fed into the VLMs. Therefore, its effectiveness is not strictly tied to the specific downstream LLMs, but rather to the shared vision encoder used to determine the semantic stability stopping condition. In Table 4, we tested DiffCAP using **various vision backbones**. The results showed that DiffCAP is insensitive to the specific encoder architecture, further confirming that DiffCAP generalizes well to VLMs that apply standard vision encoders.
>
> ---
>
> **Answer to W2:**
>
> We agree that preserving semantic details while removing adversarial perturbations is the central challenge of purification methods. DiffCAP is designed to address this by dynamically calculating the minimal diffusion time required for individual input, thereby **avoiding the over-purification** (information loss) and **under-purification** (inadequate robustness) often caused by fixed-step diffusion methods. This discussion is contained in Appendix D.4, we also detailed the new quantitative results in our response to **Q2**.

---

> ### Author Response · Authors · 2025-11-22
>
> **Answer to Q1:**
>
> In this rebuttal, we supply with experiments on the newer and larger VLM **MiniGPT-4**, which aligns a frozen visual encoder with the powerful Vicuna, enabling highly detailed image descriptions and complex reasoning. We evaluated DiffCAP against the **AttackVLM** (Zhao et al., 2023), which proposed more advanced **transfer-based** and **query-based adversarial attacks**, specifically designed to mislead VLMs into generating specific target captions, rather than plain untargeted degradation.
>
> We strictly follow their evaluation protocol, where the clean images are sampled from the ImageNet1K dataset and a target text is randomly selected from the COCO captions for each clean image. We report the **CLIP score** between the generated responses of input images and predefined targeted texts, as computed by various CLIP text encoders and their average. The prompt is fixed as “what is the content of this image?”. Pretrained CLIP encoders (ViT-B/32) are used as surrogate models for attacks.
>
> |  | RN50 | RN101 | ViT-B/32 | ViT-B/16 | ViT-L/14 | Ensemble |
> | --- | --- | --- | --- | --- | --- | --- |
> | clean (Baseline)   | 0.383 | 0.619 | 0.375 | 0.361 | 0.356 | 0.419 |
> | MF-ii (No Defense)   | 0.689 | 0.784 | 0.772 | 0.732 | 0.674 | 0.730 |
> | MF-ii (DiffCAP)  | **0.470** | **0.505** | **0.481** | **0.459** | **0.411** | **0.465** |
> | MF-ii + MF-tt (No Defense)   | 0.756 | 0.752 | 0.787 | 0.772 | 0.750 | 0.763 |
> | MF-ii + MF-tt (DiffCAP)   | **0.405** | **0.528** | **0.439** | **0.422** | **0.380** | **0.435** |
>
> As shown in the table, the 'No Defense' settings yield high CLIP scores, indicating that the VLM was successfully manipulated into generating the target text. However, DiffCAP maintains its effectiveness under MF-ii (image-to-image transfer) and MF-ii + MF-tt (joint image-text query) attacks, reducing the CLIP scores comparable to, and in some cases lower than, the clean baseline. The results verify the **generalizability of DiffCAP for both state-of-the-art VLM architectures and attack strategies**.
>
> ---
>
> **Answer to Q2:**
>
> We assessed the perceptual quality and semantic preservation of the purified images compare to their clean counterparts on the Image Captioning task under $\ell_{\infty}^{4/255}$ adversarial perturbations. We employed three standard metrics: PSNR (Peak Signal-to-Noise Ratio) measures **pixel-level signal fidelity**; LPIPS (Learned Perceptual Image Patch Similarity) measures **perceptual distance** using deep features, aligning closely with **human visual perception**; CLIP Score measures the **semantic consistency**.
>
> |  | PSNR↑ (DiffPure) | PSNR↑ (DiffCAP) | LPIPS↓ (DiffPure) | LPIPS↓ (DiffCAP) | CLIP↑ (DiffPure) | CLIP↑ (DiffCAP) |
> | --- | --- | --- | --- | --- | --- | --- |
> | OF-COCO | 24.27 dB | **29.03 dB** | 0.296 | **0.160** | 0.871 | **0.930** |
> | OF-Flicker30k | 23.44 dB | **28.21 dB** | 0.298 | **0.151** | 0.865 | **0.930** |
> | LLaVA-COCO | 25.41 dB | **29.76 dB** | 0.304 | **0.200** | 0.849 | **0.893** |
> | LLaVA-Flicker30k | 24.59 dB | **29.08 dB** | 0.306 | **0.194** | 0.837 | **0.884** |
>
> The comparison results between DiffCAP and DiffPure are presented above, DiffCAP significantly outperforms DiffPure across all three metrics with different VLMs and datasets. The higher PSNR and lower LPIPS indicate that DiffCAP **preserves fine-grained visual details** and **reduces perceptual artifacts** effectively. The superior CLIP Scores further validate our method's **high semantic integrity**, establishing a new state-of-the-art balance between robustness and faithfulness among purification methods.

---

### Author Response · Authors · 2025-11-26
**Unified Response - Part I**

We thank the reviewers for their patience and valuable suggestions, which have helped us improve the clarity and completeness of our analysis regarding the model's critical hyperparameters. At this stage, we have addressed all points raised by the four reviewers and will update the manuscript before Dec. 1. If this and our previous rebuttals have addressed your concerns, we would be grateful if you could consider raising your ratings accordingly. The authors stand ready to engage in further discussion should you have additional comments.

---

> ### Lightweight One-Time Calibration:

We agree that the calibration step can optimize performance of DiffCAP in a new domain. However, compared with widely adopted adversarial training strategies, **this requirement is minimal in both data and compute**. In realistic deployments such as autonomous driving, access to a small “calibration set” is a standard availability assumption, whereas collecting and maintaining large-scale clean and adversarial examples for VLM fine-tuning is the actual barrier—typically orders of magnitude more than what Alg. 2 needs. In our experiments, $100$ clean–adversarial image pairs are sufficient for calibration within a few seconds. By contrast, **adversarial fine-tuning of large VLMs often demands tens of GPU-hours**. This makes our approach substantially more practical in real-world settings.

> ### The Variance of the Threshold:

The number of images in the “calibration set” for Alg. 2 has a faint impact on threshold calculation. When we vary the subset size of image pairs to $100$, $200$, $300$, and use three different random seeds, the resulting $\tau$ remains stable at $0.958\pm0.003$.

>  ### The Threshold for Out-of-distribution (OOD):

We acknowledge that severe distribution shifts naturally degrade VLM performance compared to in-domain natural images. For example, VLMs can underperform simple CNN classifiers on datasets like MNIST. However, our empirical results demonstrate that $\tau$ is highly robust, and **Alg. 2 serves as an optimizer rather than a strict dependency**.

**Robustness of the Default Threshold (Evidence from MNIST)**. As detailed in Appx. D. 4 (Tab. 8), we tested DiffCAP on MNIST, which represents a severe domain shift from the natural training data. Using the default $\tau=0.96$ yields an accuracy of $79.6\%$, surpassing the clean baseline of $75.2\%$ under $\ell_\infty^{4/255}$ attack. Calibrating specifically for MNIST via Alg. 2 yields $\tau=0.97$, improving accuracy marginally to $80.9\%$. This validates that the default threshold is robust for OOD data. **The calibration provides a minor performance boost typical of hyperparameter tuning but is not required to achieve SOTA defense.**

**Impact of Semantic Density (New Experiments on ImageNet-R/S)**. To further analyze how semantic complexity affects the acquisition of $\tau$, we conducted additional experiments on two datasets with **distinct styles**: ImageNet-R (art, cartoons, and other diverse styles representing *rich* semantics) and ImageNet-S (black and white outlines representing *sparse* semantics). We **compressed resolutions** and crafted **long-tail categories** to simulate the requested OOD conditions. Table below shows the Top-1 accuracy of zero-shot classification with CLIP under $\ell_\infty^{4/255}$ attack.

||ImageNet-R(endition)|ImageNet-S(ketch)|
|-|-|-|
|Clean|75.5|58.5|
|No defense|0.0|0.1|
|DiffPure|63.5|50.1|
|DiffCAP ($\tau=0.95$)|**69.2**|51.9|
|DiffCAP ($\tau=0.96$)|67.6|52.5|
|DiffCAP ($\tau=0.97$)|66.7|**53.1**|

Alg. 2 calibrates $\tau=0.95$ for ImageNet-R and $\tau=0.97$ for ImageNet-S. This aligns with our intuition: **rich semantics are easier to stabilize, necessitating a slightly lower threshold, whereas sparse semantics reach the recovery region harder**. The former contains abundant textural and color cues that facilitate faster feature reconstruction by the diffusion model, and the latter is more sensitive to noise interference. Despite these differences, the performance variance across the $\tau\in[0.95, 0.97]$ is inapparent. Even using a sub-optimal threshold, DiffCAP consistently outperforms the DiffPure.

**Threshold is not brittle to severe domain shifts in terms of both measurement and deployment.** Users can safely tune within the $0.95-0.97$ "safety belt" without the risk of losing SOTA performance, although Alg. 2 offers a sound and quick hyperparameter search.

> ### The Threshold for Different Tasks:

The calculation of $\tau$ by Alg. 2 is based on the semantic stability of image embeddings and is therefore **task-agnostic**. Here we empirically verify whether the calibrated threshold is still optimal across different downstream tasks. As shown in the main paper, for the image captioning task on COCO with OF under $\ell_\infty^{2/255}$ attack, the calibrated $\tau = 0.96$ delivers the overall highest CIDEr scores across various diffusion step sizes $\Delta t$, reflecting the effectiveness of Alg. 2.

---

> ### Author Response · Authors · 2025-11-26
> **Unified Response - Part II**
>
> To demonstrate **task transferability** of $\tau$, we conducted an analogous ablation for the visual question answering task on VQAv2 with LLaVA under $\ell_\infty^{4/255}$ attack. The tables below report the quantitative results, which indicate that $\tau=0.96$ achieves the highest accuracy at the default step size $\Delta t=0.010$ while maintaining the efficiency. This is consistent with our observation in Fig. 2. We will include the corresponding visualization similar to Fig. 2 in the Appendix of the revised manuscript.
>
> |VQA Accuracy|$\Delta t=0.005$|$\Delta t=0.010$|$\Delta t=0.015$|$\Delta t=0.020$|
> |-|-|-|-|-|
> |$\tau=0.95$|62.8|64.0|65.8|65.6|
> |$\tau=0.96$|64.0|69.6|67.2|64.6|
> |$\tau=0.97$|65.6|65.0|62.0|60.6|
> |$\tau=0.98$|64.8|60.6|56.0|53.6|
>
> |Runtime (s/img)|$\Delta t=0.005$|$\Delta t=0.010$|$\Delta t=0.015$|$\Delta t=0.020$|
> |-|-|-|-|-|
> |$\tau=0.95$|0.39|0.88|1.72|2.74|
> |$\tau=0.96$|0.43|1.13|2.44|3.87|
> |$\tau=0.97$|0.49|1.62|3.14|5.12|
> |$\tau=0.98$|1.25|4.47|6.40|9.11|
>
> > ### The Threshold for Various CLIP:
>
> Alg. 2 employs a vision encoder to quantify semantic stability. To assess whether the encoder architecture biases the threshold calibration, we applied Alg. 2 with several CLIP variants. The calibrated $\tau$ values are listed below, clustering around $0.96$:
>
> ||RN50|RN101|ViT-B/32|ViT-B/16|ViT-L/14|
> |-|-|-|-|-|-|
> |$\tau$|0.964|0.967|0.958|0.956|0.962|
>
> We posit that for natural images, **unless the encoder architecture is fundamentally substituted** (e.g., departing from the contrastive learning paradigm), **the semantic threshold is principally governed by the underlying data distribution rather than the specific vision backbone**. This clue retrospectively explains the results in Tab. 4 of the original paper, where we revealed that $\tau=0.96$ keeps effective when the DiffCAP vision encoder is replaced by other CLIP variants. In summary, DiffCAP is largely insensitive to the vision backbone used for either calibration or purification.
>
> > ### The Step Size and Robustness-Fidelity:
>
> The trade-off is primarily influenced by the diffusion depth: deeper diffusion enhances robustness but risks losing information, while shallower diffusion preserves details but leaves residual adversarial noise. In DiffCAP, the diffusion depth is dynamically determined by the coupling between the $\Delta t$ and $\tau$:
>
> - Larger $\Delta t$ "jumps farther" in the embedding space between steps. If combined with a high $\tau$, i.e. a strict stability requirement, the forward diffusion may **overshoot the recovery region**, leading to redundant steps and over-purification.
> - Smaller $\Delta t$ induces finer granularity. If combined with a low $\tau$, the stability condition can be triggered too early, causing **premature stopping** and under-purification.
>
> Therefore, an optimal trade-off requires balancing $\Delta t$ and $\tau$, **avoiding configurations where they are simultaneously too large or too small**. Fig. 2 and the results discussed above manifest that, across different tasks, datasets, attack magnitudes, and VLMs, there exists a relatively safe range of $\Delta t$. With $\tau=0.96$ fixed, $\Delta t\in[0.05, 0.15]$ consistently outperforms DiffPure, where $\Delta t=0.10$ is set as a reliable and efficient default.
>
> > ### The Justification of Scheduler:
>
> We employ a linear noise schedule in DiffCAP for theoretical guarantee, alignment with diffusion dynamics, and empirical performance.
>
> **Our theoretical derivations are formulated under the linear precondition**. Regardless of whether the noise injection follows a linear or non-linear schedule, the cumulative noise will eventually push the image into the certified recovery region. With a sufficiently small step size ($\Delta t\approx0.01$), the specific trajectory only marginally shifts the precise timestamp at which this region is entered but does not alter the fundamental semantic convergence behavior.
>
> **DiffCAP operates in conjunction with a pre-trained diffusion model for the reverse denoising step**, where linear $\beta$ schedule is adopted. To support **DiffCAP is insensitive to the exact choice of schedule**, we compare our default Linear schedule against a cosine ($\Delta t'=\Delta t\cdot\cos(\frac{\pi i}{2N})$) and an exponential decay ($\Delta t'=\Delta t\cdot0.9^{i}$) schedule, where $i$ denotes the step index and $N$ is the total number of steps. As presented below, the alternative schedules exhibit no material performance differences:
>
> ||Linear|Cosine|Exponential Decay|
> |-|-|-|-|
> |CIDEr for OF-COCO-$\ell_\infty^{2/255}$|79.3|79.2|79.2|
> |VQA Acc. (%) for LLaVA-VQAv2-$\ell_\infty^{4/255}$|68.5|68.0|68.1|
>
> While the current linear schedule is well-founded and effective, we appreciate the reviewer’s insight regarding the scheduling strategy. **Developing a fully learnable noise controller is a promising direction for future extensions of DiffCAP.** However, we believe the current framework already furnishes a remarkable contribution to the field.

---

### Author Response · Authors · 2025-12-03
**Cover Letter of Rebuttal Summarization to Area Chair**

Dear Area Chair,

We sincerely thank your dedication assessing our work in this hard time. With the original responses and updated manuscript, we hope this letter can summarize our improvements and resolutions for all concerns.
# Contribution
We appreciate the reviewers for recognizing:
- **Reviewer b8i8 (Subscore: 10)**: "rigorous theoretical contributions", "addresses a key limitation in prior work", "strong robustness even against adaptive white-box attacks".
- **Reviewer FdbB (Subscore: 10)**: "clear motivation vs. fixed-time purification", "theoretical grounding", "simple, plug-in pipeline".
- **Reviewer gzhU (Subscore: 9)**: "significantly outperforms existing defense", "strong theoretical foundation", "well-written, well-structured".
- **Reviewer s6Go (Subscore: 7)**: "theoretical proofs guarantee", "extensive experiments validate the effectiveness", "elegant and effective per-input adaptivity".
# Response to Reviewer b8i8:
- **Model Generalizability (W1/Q1)**: We explained that OpenFlamingo and LLaVA 1.5 represent the two dominant VLM paradigms. We added new evaluations on MiniGPT 4 against AttackVLM (Appx. D.4), confirming DiffCAP’s effectiveness across more complex architectures and attacks.
- **Robustness-Fidelity (W2/Q2)**: We addressed the concern about potential distortion by providing new quantitative results with PSNR, LPIPS, CLIP Score (Appx. D.6). DiffCAP significantly outperforms DiffPure in preserving both perceptual quality and semantic integrity.
# Response to Reviewer FdbB:
- **Computational Overhead (W2/Q2)**: We reported inference latency and memory usage of DiffCAP and baselines (Appx. D.7). The time breakdown of core steps in DiffCAP is already in original paper (Line 470-478).
- **White-box Adversary (W3/Q3)**: We detailed the implementation where BPDA has the full knowledge of the DiffCAP, including the CLIP, the stopping rule, and the threshold. We supplied with a sanity check to validate that adaptive attacker indeed exploits the DiffCAP's gradient when crafting adversarial examples.
# Response to Reviewer gzhU:
- **Theoretical Coverage (W1)**: We justified the applicability of our theoretical analysis to general VLM tasks (Appx. C.4), as VLM generation is conditioned on the vision embedding, and discriminative VLM tasks can be formulated as probability distributions over tokens, fitting our classifier assumption.
- **Large Perturbations (W2)**: We conducted experiments under $\ell_\infty^{16/255}$ attack (Appx. D.3). DiffCAP maintained superior robustness compared to other defenses.
- **Other VLMs (W3)**: We evaluated on MiniGPT 4 against AttackVLM (Appx. D.4), together with experiments on MiniGPT 4 under visual jailbreakings (Appx. D.5), to demonstrate the transferability of DiffCAP to more sophisticated models and threats.
# Response to Reviewer s6Go:
- **Novelty (W1/Q4)**: We articulated that DiffCAP drives DiffPure from the static, empirical purification logic to the difficulty-adaptive, theorem-grounded semantic restoration with SOTA efficacy-efficiency-fidelity balance.
- **Adaptive Attack Implementation (W3/Q1)**: We described in detail (Appx. D.1) where BPDA approximates the backward gradient of DiffCAP as an identity matrix to bypass the gradient obfuscation and instability, and EOT estimates the expected gradients by averaging over multiple stochastic forward passes.
- **Backbone Vulnerability (W4/Q3)**: We resolved the CLIP ViT-B/32 reliance issue by testing DiffCAP against CLIP-as-surrogate attacks (AttackVLM) on MiniGPT 4 (Appx. D.4), verifying that our semantic stability criterion is robust to backbone-aware exploitation.
# Unified Response:
We performed comprehensive ablations on key hyperparameters of DiffCAP in Appx. E.
- **Calibration Dependency (b8i8 W3)**: We clarified that the $\tau$ calibration is a negligible one-time cost compared to computationally expensive adversarial training methods. The default $\tau=0.96$ is highly robust, making calibration optional for deployment.
- (**FdbB W1/W4/Q1, gzhU Q1, s6Go W2/Q2**) We studied how **out-of-distribution** influences both the acquisition and application of **threshold** in Appx. E.2. And:
- (**FdbB W1/W4/Q1, gzhU Q1**) The **threshold** for different **tasks** in Appx. E.3.
- (**FdbB W1, gzhU Q1, s6Go Q2**) The **threshold** for **CLIP variants** in Appx. E.4.
- (**FdbB W4, gzhU Q1**) The **threshold** for calibration **subset sizes** and **random seeds** in Appx. E.1.
- (**FdbB W1/Q3**) The **step size** for **robustness-fidelity** trade-off in Appx. E.5.
- (**FdbB W1/Q3**) The **noise schedule** discussion in Appx. E.6.
# Summary
In this rebuttal, we have remarkably enriched the paper with **7 new tables**, **2 new figures**, and **5 new pages** of analysis. We have also attached a **confidential message** to bring a review-related concern to your attention. We believe DiffCAP stands as a valuable and insightful contribution to the VLM safety community.

Best regards,

The Authors of Submission 2637

---

### Meta-Review · Area_Chair_BfBT · 2026-01-07

**Summary:**

The reviewers generally appreciated the paper’s theoretical grounding and methodological clarity. However, they converged on a shared concern that the contribution appears incremental relative to prior diffusion-based purification approaches such as DiffPure. Central issues included limited demonstration of novelty, insufficient validation across diverse vision-language model architectures, and an underdeveloped discussion of the reliance on and practical cost of calibrating the adaptive stopping threshold. Although the authors supplied additional experiments during rebuttal, these core reservations were not fully dispelled, leading to an overall assessment that falls short of the acceptance bar.

**Reviewer Concerns:**

The authors responded to all weaknesses and questions raised by the reviewers in their rebuttal.

**Reviewer Scores:**

Reviewer b8i8 initially gave a score of 6 and would likely raise it to 7 or 8, as their primary concerns were well resolved.
Reviewer FdbB gave a 4 but might adjust it slightly upward to 5. However, their review contains multiple factual errors and logical inconsistencies, which substantially reduce its credibility and should limit its influence on the final decision.
Reviewer gzhU started at 4 and would probably maintain their score.
Reviewer s6Go began with a 6 and may increase it modestly to 7, though their skepticism about the work's novelty remains only partially alleviated.

---

### Decision · Program_Chairs · 2026-01-26

Reject